# High-throughput library transgenesis in *Caenorhabditis elegans* via Transgenic Arrays Resulting in Diversity of Integrated Sequences (TARDIS)

**Zachary C Stevenson[1], Megan J Moerdyk-Schauwecker[1], Stephen A Banse[1], Dhaval S Patel[2,3,†], Hang Lu[2,3], Patrick C Phillips[1]\***

[1]Institute of Ecology and Evolution, University of Oregon, Eugene, United States; [2]School of Chemical & Biomolecular Engineering, Georgia Institute of Technology, Atlanta, United States; [3]Petit Institute for Bioengineering and Bioscience, Georgia Institute of Technology, Atlanta, United States

**\*For correspondence:**
pphil@uoregon.edu

**Present address:** †NemaLife Inc, Lubbock, United States

**Abstract** High-throughput transgenesis using synthetic DNA libraries is a powerful method for systematically exploring genetic function. Diverse synthesized libraries have been used for protein engineering, identification of protein–protein interactions, characterization of promoter libraries, developmental and evolutionary lineage tracking, and various other exploratory assays. However, the need for library transgenesis has effectively restricted these approaches to single-cell models. Here, we present Transgenic Arrays Resulting in Diversity of Integrated Sequences (TARDIS), a simple yet powerful approach to large-scale transgenesis that overcomes typical limitations encountered in multicellular systems. TARDIS splits the transgenesis process into a two-step process: creation of individuals carrying experimentally introduced sequence libraries, followed by inducible extraction and integration of individual sequences/library components from the larger library cassette into engineered genomic sites. Thus, transformation of a single individual, followed by lineage expansion and functional transgenesis, gives rise to thousands of genetically unique transgenic individuals. We demonstrate the power of this system using engineered, split selectable TARDIS sites in *Caenorhabditis elegans* to generate (1) a large set of individually barcoded lineages and (2) transcriptional reporter lines from predefined promoter libraries. We find that this approach increases transformation yields up to approximately 1000-fold over current single-step methods. While we demonstrate the utility of TARDIS using *C. elegans*, in principle the process is adaptable to any system where experimentally generated genomic loci landing pads and diverse, heritable DNA elements can be generated.

## eLife assessment

This manuscript provides a description of an approach for efficiently integrating diverse libraries into the *C. elegans* genome and tools that enable researchers to use the method. It is a **valuable** contribution for researchers carrying out experiments that would benefit from easy generation of such libraries, and the data for the effectiveness of the method is **solid**. The advantages of this approach in terms of ease and effectiveness relative to others with similar aims will emerge as they are put to more general use in addressing biological problems.

**eLife digest** Transgenesis – the ability to insert foreign genetic material (known as transgenes) in to the genome of an organism – has revolutionized biological research. This approach has made it possible for scientists to study the role of specific genes and to produce animal models which mimic aspects of human diseases.

For transgenes to be maintained and passed down to future generations, they must be introduced into germ cells which will go on to form the egg and sperm of the organism. However, despite advances in genetic engineering, this process (called 'specific transgenesis') is still laborious and time-consuming, and limits researchers to working with only a small number of known DNA sequences at a time.

In contrast, 'exploratory transgenesis' – where dozens of transgenes from a library of DNA sequences are introduced simultaneously into multiple individuals – is more efficient and allows for more large-scale experiments. However, this approach can only be done with single-celled organisms like bacteria, and remains virtually impossible in laboratory animals like worms or mice.

Stevenson et al. therefore set out to boost the efficiency of exploratory transgenesis in a commonly used laboratory animal, the roundworm *Caenorhabditis elegans*. To do this, they used the 'library' principle of exploratory transgenesis in order to develop a new resource called TARDIS (short for, Transgenic Arrays Resulting in Diversity of Integrated Sequences).

First, Stevenson et al. genetically engineered worms to carry a 'landing site' for foreign DNA. Next, a library of transgenes and a mechanism which cuts pieces of DNA and pastes them into the landing site were introduced into the germ cells of these worms using traditional methods. The worms were then bred to generate a large population of offspring that had inherited this array of foreign DNA sequences. Finally, the 'cut and paste' mechanism was switched on and a random transgene was inserted into the landing site in the genome. This resulted in thousands of worms which each had a unique genetic modification that can be passed on to future generations.

These results show for the first time that larger-scale transgenesis experiments are possible in multi-cellular animals. In the future, Stevenson et al. hope that TARDIS can be adapted to different organisms and allow researchers to carry out experiments that were not previously possible.

## Introduction

Transgenesis, which is the specific and heritable introduction of foreign DNA into genomes, has been a central tool for functional analysis and genetic engineering for nearly 40 y. The power of transgenesis is due in part to the wide variety of assays and techniques that are built upon controlled introduction of novel DNA sequences into a native genome. While there are many uses for transgenesis, in practice most can be grouped into those inserting a small number of known sequences (specific transgenesis) and those introducing many sequence variants from experimental libraries (exploratory transgenesis). While the ability to perform specific transgenesis has become a de facto requirement for all model organisms, exploratory transgenesis remains effectively limited to single-cell models (both prokaryotic and eukaryotic) because of biological limitations generated by inheritance in multicellular organisms. In single-cell models, high-throughput transgenesis has been used for exploratory sampling of sequence space using protein interaction libraries (*Joung et al., 2000*), barcode-lineage tracking libraries (*Levy et al., 2015*; *Nguyen Ba et al., 2019*), directed evolution (*Packer and Liu, 2015*), synthetic promoter library screens (*Wu et al., 2019*), and mutagenesis screens (*Bock et al., 2022*; *Erwood et al., 2022*; *Kim et al., 2022*; *Sánchez-Rivera et al., 2022*). Despite the usefulness of such experiments in single-celled systems, either in microorganisms or in cell culture, increasing transgenic throughput in multicellular models holds the potential to expand the impact of exploratory transgenesis in functional domains, such as inter-tissue signaling, neuronal health, and animal behavior, that are dependent on multicellular interactions and therefore difficult to replicate in single-cell models.

Exploratory transgenesis in single-cell models has been facilitated by the availability of in vitro-generated DNA libraries, selectable markers, plasmids, in vivo homologous recombination, and most importantly, the ability to massively parallelize transgenesis using microbial transformation or eukaryotic cell transfection/transduction. Currently, there is no practical means to make populations

of uniquely transgenic individuals from sequence libraries at a similar scale in animal systems due to the Weismann barrier (*Weismann, 1893*): the split between soma and germline. The requirement that the germline be accessible and editable has forced animal systems into a transgenic bottleneck compared to single-cell systems because it is very difficult to introduce exogenous DNA directly into the germline in a high-throughput manner, relying instead on injection, bombardment, or some other physical intervention. This low-throughput limitation in animals dramatically reduces the sequence diversity that can be sampled, effectively preventing large-scale exploratory experiments from being performed. Attempts have been made to parallelize transgenic creation in multicellular model organisms, for example, the development of Brainbow (*Livet et al., 2007*; *Weissman and Pan, 2015*), ifgMosaic analysis (*Pontes-Quero et al., 2017*), P[acman] libraries in *Drosophila* (*Venken et al., 2009*), and multiple types of transformation in plants (*Ismagul et al., 2018*; *Xu et al., 2022*). In *Caenorhabditis elegans*, CRISPR technology combined with custom engineered sites within the genome ('landing pads') has facilitated the generation of single-copy integrations (*Malaiwong et al., 2023*; *Nonet, 2020*; *Nonet, 2021*; *Silva-García et al., 2019*; *Stevenson et al., 2020*; *Vicencio et al., 2019*), and attempts have been made to multiplex transgenesis using traditional integration methods in conjunction with specialized landing pad systems (*Gilleland et al., 2015* ; *Kaymak et al., 2016*; *Mouridi et al., 2022*; *Radman et al., 2013*). While these efforts have increased throughput over standard single-copy integration methods, throughput still remains too low for effective exploratory transgenesis, and in some cases requires significant additional labor, cost, equipment, and/or expertise.

Here, we present 'Transgenic Arrays Resulting in Diversity of Integrated Sequences' (TARDIS) (*Stevenson et al., 2021*), a simple yet powerful alternative to traditional single-copy transgenesis. TARDIS greatly expands throughput by explicitly separating and reordering of the conceptual steps of transgenesis (*Figure 1*). To increase throughput, TARDIS begins with an in vitro-generated DNA sequence library that is introduced into germ cells via traditional low-throughput methods (i.e., germline transformation, *Figure 1*). While traditional transgenesis typically couples the physical introduction of DNA into cells with the integration of a selected sequence from the original library, the DNA sequences in TARDIS are designed to be incorporated in large numbers into diverse, heritable sub-libraries (TARDIS libraries), rather than be directly integrated into the desired genomic locus. In addition to the sequence library, a functioning selectable marker is also included to stabilize the inheritance of the TARDIS library over generations. These TARDIS libraries function to create 'meta-ploidy' – expanding the total number of alleles available for inheritance, essentially making the worm genetically 'bigger on the inside.' TARDIS library-bearing animals are then allowed to propagate under selection to generate a large population of TARDIS library carriers. After population expansion, genome integration of a single-sequence unit is performed by inducing a double-strand break at a genetically engineered landing pad. This landing pad is designed to both integrate a sequence unit and act as a second selectable marker. We chose *C. elegans* to validate the TARDIS approach because *C. elegans* naturally form extrachromosomal arrays that can be several megabases in size (*Carlton et al., 2022*; *Lin et al., 2021*; *Mello et al., 1991*; *Stinchcomb et al., 1985*) from injected DNA, which simplifies the generation of heritable 'TARDIS library arrays' (TLA) that encompass significant sequence diversity.

We demonstrate the functionality of TARDIS for two use cases: unique animal barcoding and promoter library transgenesis. Barcoding has been widely adopted in microbial systems for evolutionary lineage tracking (*Jahn et al., 2018*; *Levy et al., 2015*; *Nguyen Ba et al., 2019*) and for developmental lineage tracking in animals (*Kebschull and Zador, 2018*; *McKenna et al., 2016*). In microbial systems, barcode libraries have relied on highly diverse randomized oligo libraries, compared to animal systems, which have relied on CRE recombinases or randomized Cas9-induced mutations. Here, we present a novel TARDIS barcoding system for an animal model that mimics the scope and diversity previously only possible using microbial systems. Our results show that large, heritable libraries containing thousands of barcodes can be created and maintained as extrachromosomal arrays. Individual sequences are selected and removed from the library upon experimental induction of Cas9 in a proportion consistent with the composition of the TLA with rare overrepresented sequences. We found that TARDIS is also compatible with the integration of large promoters and can be used to simultaneously integrate promoters into multiple genomic locations, providing a tool for multiple insertions at defined locations across the genome. While we demonstrate the system's advantages in *C. elegans*, in principle, the system is adaptable for any situation where the

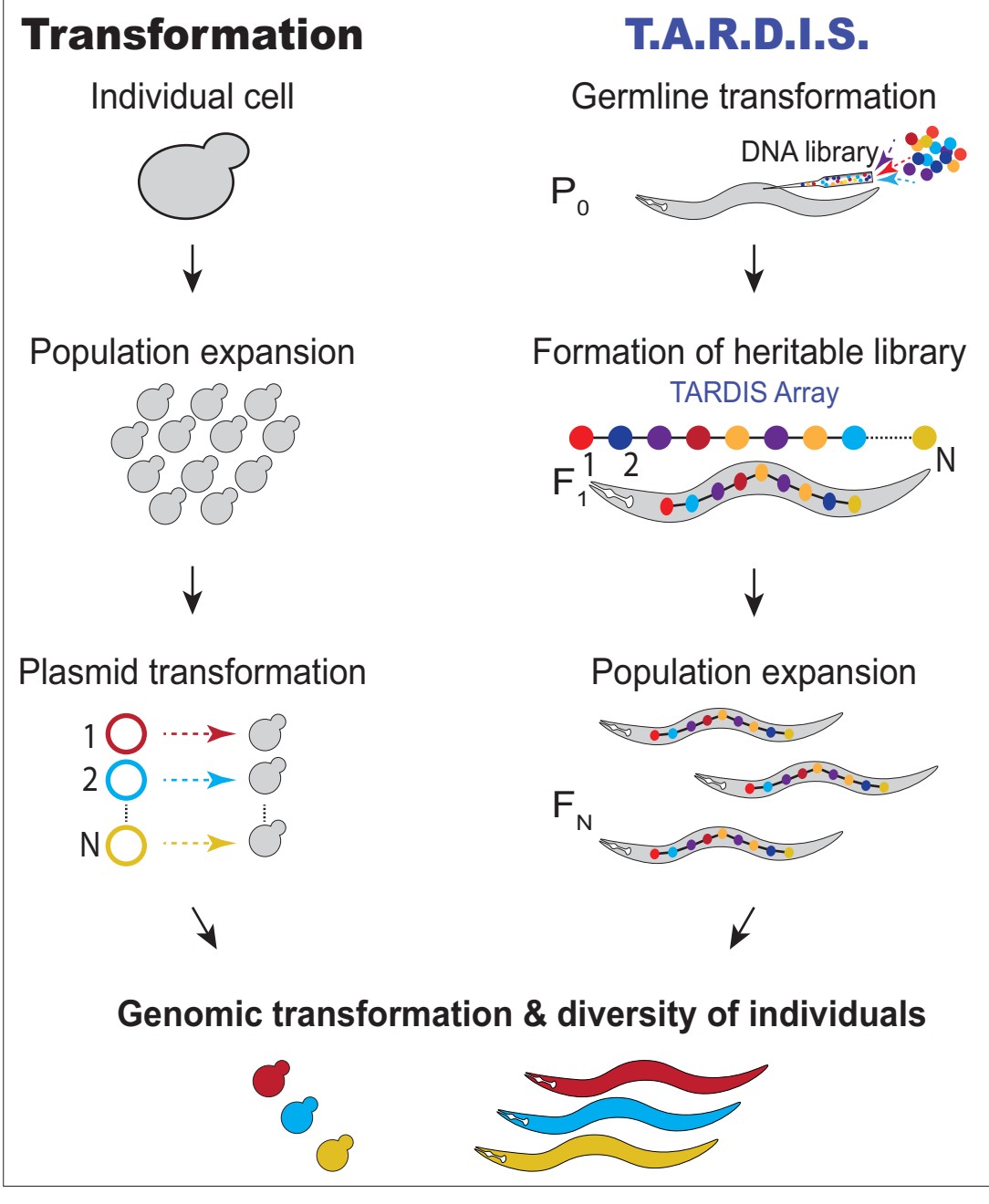

**Figure 1.** Transformation compared to Transgenic Arrays Resulting in Diversity of Integrated Sequences (TARDIS). For transformation, a large population of cells are individually transformed with a DNA library, resulting in a diverse population of individuals. TARDIS achieves a diversity of individuals by splitting transgenesis into two separate processes: (1) the introduction of a diverse library, which is formed into a TARDIS library array, passed down to future generations and thus replicated; and (2) an event that triggers the integration a sequence from the library at random, resulting in a diversity of integrated sequences.

sequences for integration can be introduced with high diversity and heritability, and where a genomic site for integration can be made or is available.

## Results

### Generation of barcode landing pad

We designed a specific landing pad for the introduction and selection of small barcode fragments from high-diversity, multiplexed barcode libraries (*Figure 2*). This landing pad was designed to be targeted by Cas9 and requires perfect integration on both the 5′ and 3′ ends of a synthetic intron for functional hygromycin B resistance. Current split selection landing pads only provide selection on one side of the double-strand break, which can result in a small percentage of incomplete integrations (*Stevenson et al., 2020*). To fully test a large library approach, the requirement of genotyping to identify correct integrations must be overcome. A split-selection, hygromycin resistance (HygR) system was chosen for its simplicity and integration-specific selection. A unique synthetic CRISPR guide RNA target sequence was created by removing coding sequence on both sides of an artificial intron, resulting in a nonfunctional HygR gene. By removing critical coding sequence on both sides of the gene, only 'perfect' integration events will result in hygromycin resistance (*Figure 2A*). The synthetic landing pad was integrated at Chromosome II: 8,420,157, which has previously been shown to be permissive for germline expression (*Dickinson et al., 2015*; *Frøkjær-Jensen et al., 2012*; *Frøkjaer-Jensen et al., 2008*).

### Generation of high-diversity donor library and TARDIS arrays

Transgenes or DNA sequences can be cloned into plasmid vectors for injections in *C. elegans*. However, the cloning process is laborious, and the plasmid vector is unnecessary for integration into an array or the genome. We sought to provide a protocol for library generation that maximized diversity and eliminated the requirement of cloning (*Figure 2B*). Oligo libraries have been used for barcoding (*Levy et al., 2015*) and for identification of promoter elements (*de Boer et al., 2020*) in yeast, but practical implementation of large synthetic libraries for transgenesis has never been performed in an animal system. We used randomized synthesized oligos to build a highly diverse library of barcodes, similar to the one described by *Levy et al., 2015*, via complexing PCR. Given randomized bases present at the 11 nucleotide positions centrally located within the barcode, our base library can yield a theoretical maximum of approximately 4.2 million sequences. Our overlap PCR approach achieves high levels of diversity with minimal 'jackpotting' – sequences with higher representation than expected (*Figure 3—figure supplement 1*). With low-coverage sequencing, we found almost 800,000 unique barcode sequences, providing a large pool of potential sequences that can be incorporated into TARDIS arrays. Only 472 sequences were overrepresented (counts greater than 50), accounting for approximately 6.7% of the total reads and only approximately 0.06% of the unique barcodes detected.

We injected our complexed barcodes and isolated individual TARDIS array lines, each containing a subset of the barcode library (*Figure 3*). Individual injected worms were singled, and we identified four arrays from three plates. Arrays 1 and 2 were identified on separate plates, and were therefore derived from independent array formation events, while array 3, profile 1 and array 3, profile 2 were both identified on the same plate. Analysis of array diversity within these lines shows, somewhat unexpectedly, that during array formation a subset of barcode sequences tended to increase in frequency (*Figure 3A and B*). Higher frequency barcodes in arrays tend to be independent of the jackpotted sequences of the injection mix as very few are represented in the set of high-frequency barcodes from the injection mix. The high-frequency barcodes also varied between arrays.

We found that array formation does not seem to favor any particular barcode sequence motif (*Figure 3C*) and that arrays can range considerably in diversity. Array 1 had 1319 unique barcode sequences, array 2 had 3001 unique barcode sequences, array 3 profile 1 had 91 unique barcode sequences, and array 3 profile 2 had 204 unique barcode sequences (*Figure 3—figure supplement 2*). Across the four arrays, we found a total of 4395 unique barcode sequences. When we compared the individual sequences incorporated during the three independent injections, we found little overlap. 96.5% (4395/4553) of the identified sequences were unique to one injection, 3.0% (136) were incorporated twice, and 0.5% (22) were recovered from all three injections. In contrast to the diversity

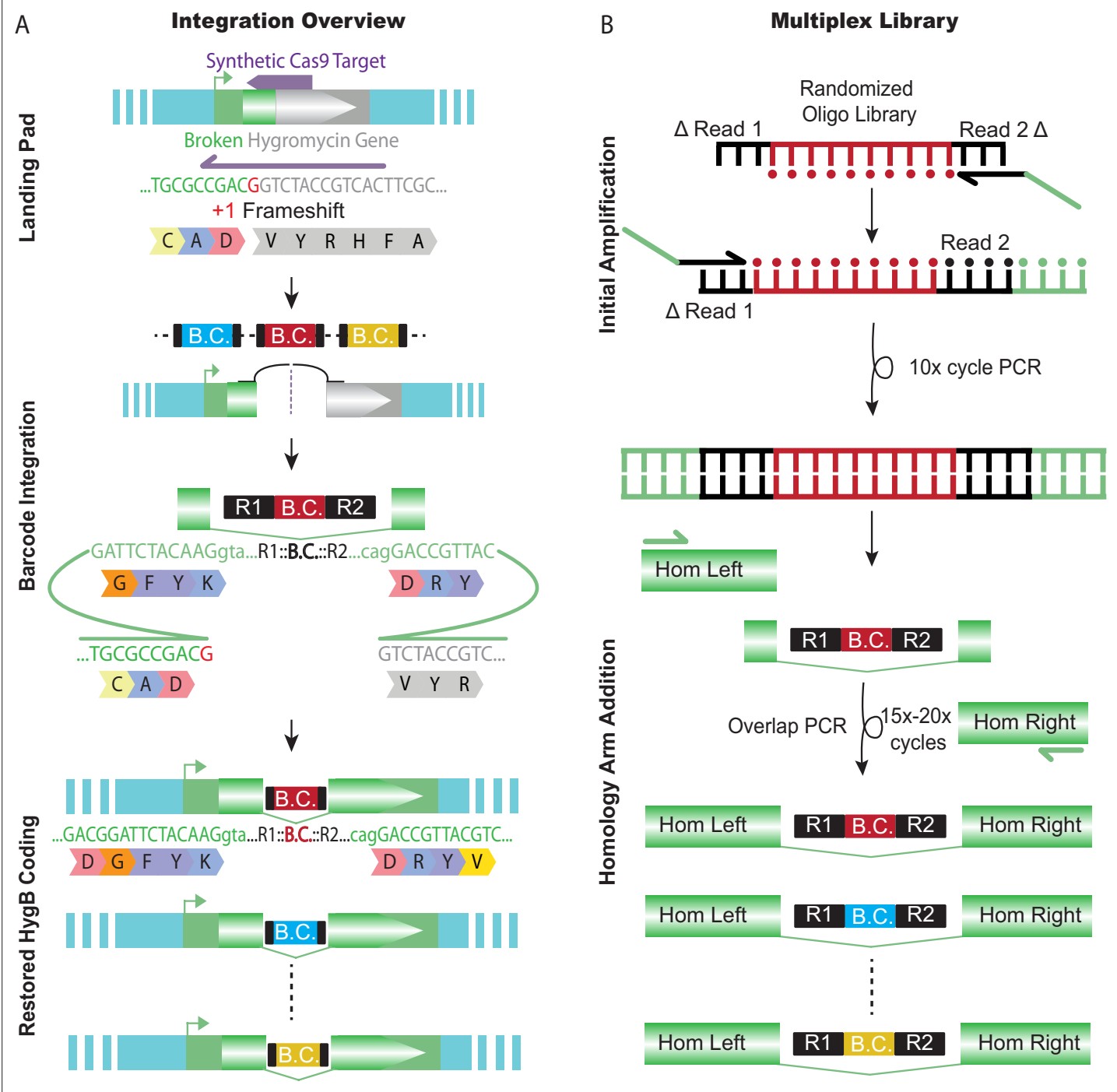

**Figure 2.** Barcode landing pad and diverse donor library. (**A**) Schematic design for the barcode landing pad and integration. A broken hygromycin resistance gene is targeted by Cas9, which repairs off the Transgenic Arrays Resulting in Diversity of Integrated Sequences (TARDIS) array, integrating a barcode and restoring the functionality of the gene. (**B**) The TARDIS multiplex library was created from a randomized oligo library, which underwent 10 cycles of PCR to make a dsDNA template. The barcode fragment was then added into a three fragment overlap PCR to add homology arms and make the final library for injection.

The online version of this article includes the following figure supplement(s) for figure 2:

**Figure supplement 1.** Schematic layout for the two separate PCR processes for identification of barcode counts in arrays (Amplicon One-Array) and integrants (Amplicon One-Integrant).

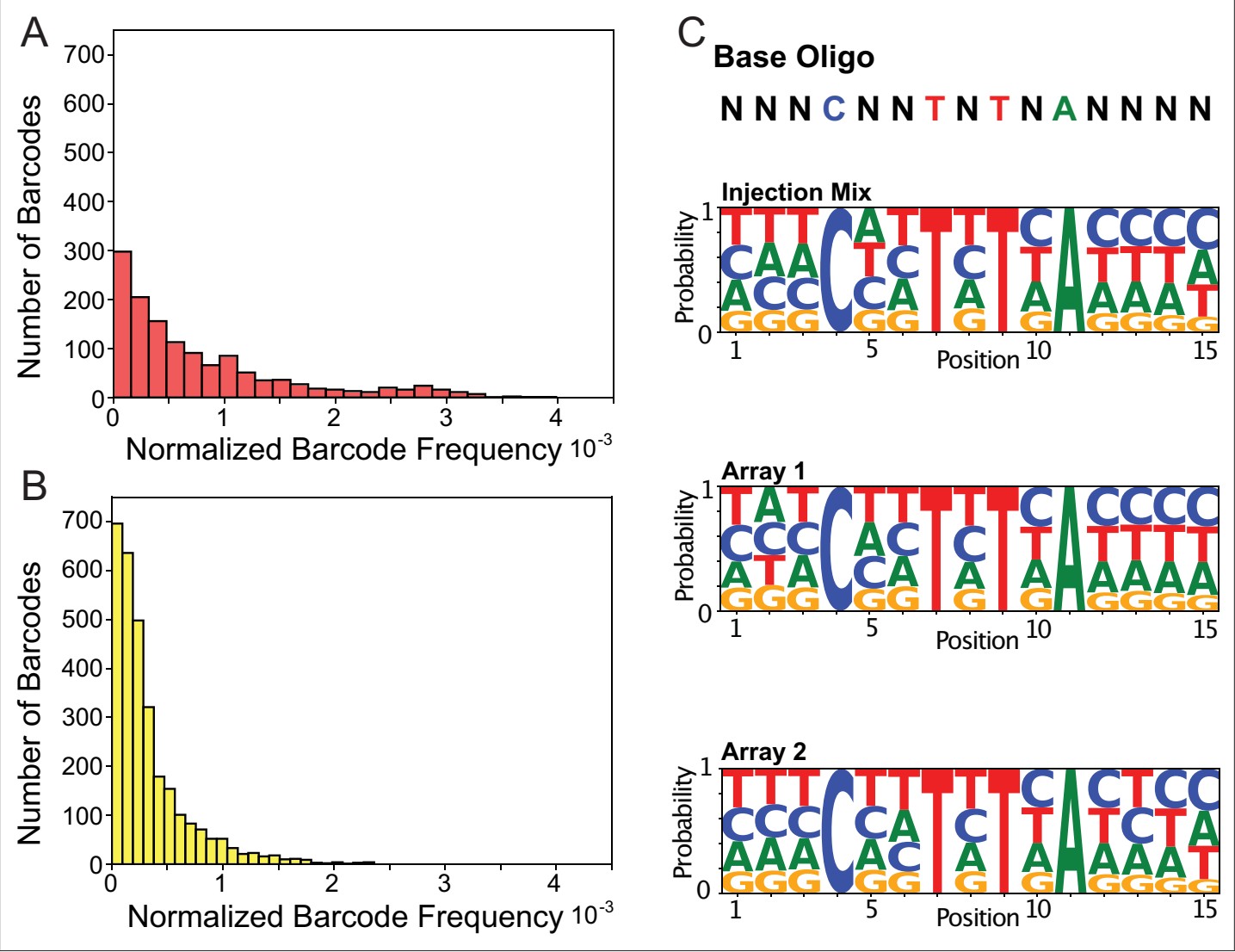

**Figure 3.** Transgenic Arrays Resulting in Diversity of Integrated Sequences (TARDIS) library arrays can contain large barcode diversity. (**A**) Frequency distribution of 1319 unique barcodes in array 1 (PX816). (**B**) Frequency distribution of the 3001 unique barcode sequences in array 2 (PX817). (**C**) Sequence logo probabilities of the 15 base pair positions of the barcodes in the injection mix, array 1 and array 2.

The online version of this article includes the following figure supplement(s) for figure 3:

**Figure supplement 1.** Barcode frequency in injection mix.

**Figure supplement 2.** Transgenic Arrays Resulting in Diversity of Integrated Sequences (TARDIS) array 3.

**Figure supplement 3.** Determination of proper count cutoff for (**A**) Transgenic Arrays Resulting in Diversity of Integrated Sequences (TARDIS) array 1 and (**B**) TARDIS array 2.

between injection events, a similar comparison of the two profiles derived from a single injection for array 3 showed considerable overlap, with 68% (62/91) of the profile 1 sequences also being present in profile 2. Overall, our results suggest our complexing PCR oligo library can produce a highly diverse library and that arrays can store a large diversity of unique sequences.

The distribution of element frequency within a given array follows a clear Poisson distribution. Arrays 1 and 2 show more diversity, with barcode frequencies more similar to one another than the two profiles isolated from array 3 (*Figure 3—figure supplement 2*). The null assumption is that the array is formed from a simple sample of the injected barcodes in equal proportions. However, arrays have been already reported to jackpot certain sequences. For example, when *Lin et al., 2021* injected fragmented DNA, they found that larger fragments were favored in the assembly. In our case, we find

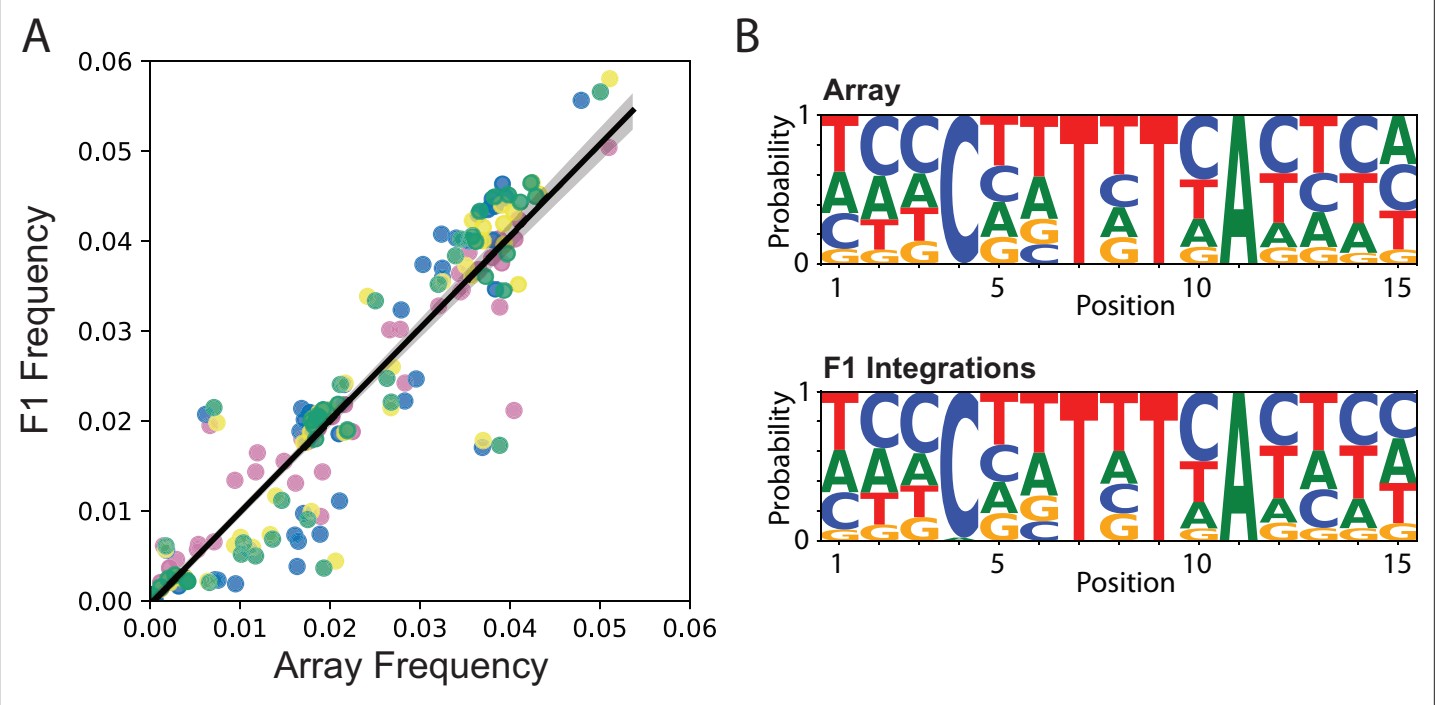

**Figure 4.** Integration frequency from Transgenic Arrays Resulting in Diversity of Integrated Sequences (TARDIS) library array to F1. (**A**) Frequency of integration from TARDIS library array to the F1, R ≈ 0.96, p≈5.7 × 10⁻¹⁵⁴. Different colors represent four biological replicates. Line shading represents 95% confidence interval. (**B**) Sequence probabilities of PX786 compared to the F1 integrations (91 unique barcodes were identified in the array and 118 in the F1s, with a five read threshold).

The online version of this article includes the following figure supplement(s) for figure 4:

**Figure supplement 1.** F1 integration events followed a consistent pattern, with replicated outlier barcode sequences.

some barcode sequences become jackpotted, despite being identical in size. A possible explanation is that early in formation, arrays are replicating sequences, possibly to reach a size threshold. Consistent with this hypothesis, arrays with higher barcode diversity had frequencies closer to one another, while arrays with lower diversity had wider frequency ranges.

## Integration from TARDIS array to F1

Our primary motivation in developing the TARDIS method was to utilize individual sequences from the TARDIS array as integrated barcodes. To assay the integration efficiency, we performed TARDIS integration on two biological replicates from a TARDIS array line (PX786) synchronized in the presence of G-418. Out of the 100 L1's per plate initially plated on antibiotic free plates, an average of 41 worms (N = 255 plates) for replicate 1 and 62 worms for replicate 2 (N = 125 plates) survived to the next day. These surviving individuals contained the array, allowing them to survive early-life G-418 exposure and generally showed fluorescent co-marker expression as well. Following heat shock to induce Cas9, replicate 1 produced 104 plates with hygromycin-resistant individuals, indicating barcode integration, and replicate 2 produced 71. These results suggest that approximately 200–300 worms need to be heat-shocked to obtain an integrated line when using 150 bp homology arms and relatively small inserts such as the barcodes. To assay the integration frequency from the array to the F1, we performed TARDIS integration on four biological replicates derived from PX786. We found that the frequency of integration for barcodes in F1 individuals was strongly correlated with the barcodes' frequency in the TLA (*Figure 4A*; $R \approx 0.96$, $p \approx 5.7 \times 10^{-154}$). Notably, there are two replicated outliers across the four biological replicates. One barcode (TTAAATTATCACATG) tended to integrate more often than would be predicted by its frequency in the array, while barcode (GCTCATTCTGACGTA) integrated less frequently than expected (*Figure 4—figure supplement 1*). In general, however, we did not observe any noticeable bias in sequence motif selection following integration (*Figure 4B*). Several individual lineages were isolated from the population with hygromycin selection, validating

functional restoration of the *HygR* gene, and three were randomly chosen for Sanger sequencing to confirm perfect barcode integration. As expected, these sequenced barcodes were also found amongst the barcode sequences of the array.

## Generation and integration of TARDIS promoter library

For testing insertion of promoter libraries via TARDIS, two separate landing pad sites utilizing split selection were engineered in chromosome II (*Figure 5A*). The first contained the 3′ portions of both the *mScarlet-I* and the *HygR* genes in opposite orientation to each other and separated by a previously validated synthetic Cas9 target (*Stevenson et al., 2020*). Similarly, the second landing pad site contained the 3′ portions of *mNeonGreen* and *Cbr-unc-119(+)* separated by the same synthetic Cas9 target, allowing both sites to be targeted by the same guide. These landing pads were engineered into an *unc-119(ed3)* background to allow for selection via rescue of the uncoordinated (Unc) phenotype. A strain containing only the split *mScarlet-I*/split *HygR* landing pad was also constructed, in which case a copy of *Cbr-unc-119(+)* was retained at the landing pad site. Repair templates contained the 5′ portion of the respective selective gene, a lox site allowing for optional removal of the selective gene after integration (by expression of Cre) and the chosen promoters in front of the 5′ portion of the respective fluorophore. The selective gene and fluorophore fragments contained >500 bp overlaps with the landing pad to facilitate homology directed repair. Correct homology directed repair at both junctions resulted in worms that were fluorescent, hygromycin resistant, and had wild-type movement.

The initial promoter library tested was composed of 13 promoters targeted to a single landing pad site with split *mScarlet-I* and split *HygR* (*Table 1*). These promoters ranged in size from 330 to 5545 bp (total repair template length of 2238–7453 bp). Seven different array lines were generated, which exhibited distinct profiles when probed by PCR as a crude measure of array composition and diversity (*Figure 5—figure supplement 1A*). Promoter-specific PCR showed these arrays to contain 2–13 of the 13 injected promoters, with a mean of 10.7 and a median of 12 (*Figure 5—figure supplement 1B*). For the selected line (PX819), 12 promoters were incorporated into the TARDIS array. From this line, approximately 200 G-418-resistant L1s (i.e., those containing the array) were plated onto each of 60 plates and then heat-shocked as L2/L3s to initiate integration. Hygromycin-resistant individuals were recovered from 59 of the 60 plates, indicating one or more integration events on each of those plates. Four individuals were singled from each of these plates, with the intent of maximizing the diversity of fluorescent profiles and analyzed by PCR to identify the integrated promoters (*Figure 5—figure supplement 1B*). Based on the banding patterns, 83 of these PCR products were sequenced with nine different promoters confirmed as integrated (*Table 1* and *Figure 5B*). This included both the smallest (*aha-1p*) and the largest promoter (*nhr-67p*) in the set. Notably, two of the three promoters that were in the array but not recovered as integrants were found to be integrated in a subsequent experiment (see below), suggesting the failure to be recovered in this case was likely due to the array composition rather than any properties of these particular promoters. For approximately half of the plates, two or more promoters were identified from the four worms chosen. Of the 83 PCR products sequenced, 5 had incorrect sequences and/or product sizes inconsistent with the promoter identified and 3 failed to prime. Additionally, several samples failed to amplify or gave a nonspecific banding pattern and likely also represent incorrect integrations.

To test whether TARDIS could be used to target multiple sites simultaneously, a second promoter library containing seven promoters targeted to each site (*ahr-1p*, *ceh-10p*, *ceh-20p*, *ceh-40p*, *ceh-43p*, *hlh-16p*, *mdl-1p*) was injected into worms containing both landing pad sites. Five plates of mixed stage worms were heat-shocked, and worms that were both hygromycin resistant and had wild-type movement were found on three of those plates. Worms that were hygromycin resistant but retained the Unc phenotype were also observed on some plates, representing individuals with integrations at a single site. For two of the plates, a single pair of integrations was observed, in both cases being *ahr-1p::mScarlet* plus *hlh-16p::mNeonGreen*. For the third plate, two different combinations were recovered: *ahr-1p::mScarlet* plus *mdl-1p::mNeonGreen* and *ceh-40p::mScarlet* plus *ceh-10p::mNeonGreen* (*Figure 5C*). While multi-site CRISPR is known to be possible (*Arribere et al., 2014*), these results suggest that TARDIS provides a unique way to engineer multiple locations using a single injection.

When transcriptional reporter lines were examined by fluorescent microscopy, expression of the fluorophores was concentrated in but not exclusive to the nucleus, consistent with the presence of nuclear localization signals (NLS) on the fluorophores. For all promoters, expression was seen in at

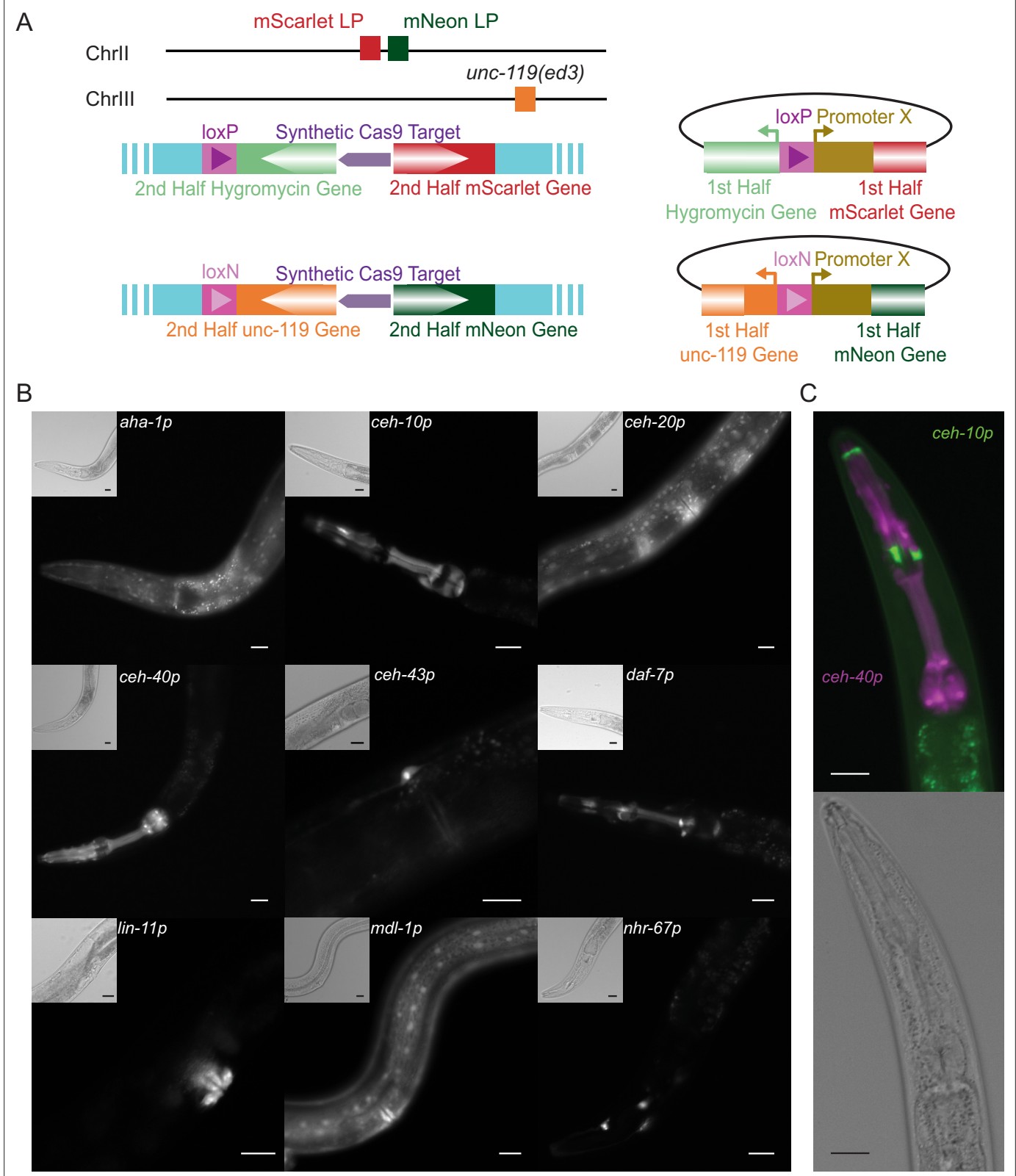

**Figure 5.** Transgenic Arrays Resulting in Diversity of Integrated Sequences (TARDIS) promoter library. (**A**) Overview of the two split landing pads and their associated promoter insertion vectors. Both the selective marker and the fluorophore expression are restored upon correct integration. (**B**) Transcriptional reporters for nine genes were recovered from a single heatshock of a single TARDIS array line (PX819). Integration was into the single mScarlet-I/HygR landing pad. Main images show mScarlet-I expression for the indicated reporter while insets show polarized image of the same region.

*Figure 5 continued on next page*

*Figure 5 continued*

(**C**) Example simultaneous, dual integration from a single TARDIS array into the double landing pad strain with PEST. ceh-10p::mNeonGreen::PEST is false-colored green and ceh-40p::mScarlet-I::PEST is false-colored magenta. All scale bars represent 20 μm.

The online version of this article includes the following source data and figure supplement(s) for figure 5:

**Figure supplement 1.** Transformation efficiency for promoter arrays.

**Figure supplement 1—source data 1.** Raw image files for results shown in *Figure 5*.

least one previously reported tissue (*Table 1*) but was absent in one or more tissues for several of the promoters. Expression of single-copy reporters is frequently more spatially restricted than that from integrated or extrachromosomal arrays (*Aljohani et al., 2020*). The differences in expression pattern may also reflect the differences in the region used as the promoter or the fact that only a single developmental stage (late L4/early adult) was examined. Overall, we find that TARDIS can be used to screen functional libraries, either individually or in combination.

## Discussion

Here, we present the first implementation of a practical approach to large-scale library transgenesis in an animal system (*Figure 1*). Building on over a half century of advancements in *C. elegans* genetics, we can now make thousands of specific, independent genomic integrations from single microinjection events that traditionally yield at most a small handful of transgenic individuals. Increasing transgenesis throughput has long been desired, and in *C. elegans* several attempts have been made to multiplex transgenic protocols. Library mosSCI and RMCE, which both introduce a multiplexed injection mixture and do indeed achieve multiple integrations (*Kaymak et al., 2016*; *Nonet, 2020*). However, just as in the case of standard mosSCI or single-donor injections for RMCE, anti-array screening, genotyping, and the direct integration of the process substantially limit the multiplex potential of these methods. One group has adopted arrays with small pools of guides coupled with heatshock-inducible Cas9 to produce randomized mutations at targeted locations (*Froehlich et al., 2021*). This protocol shares similarities with TARDIS, in that diverse arrays are coupled with inducible Cas9. However, the focus of

**Table 1.** Characteristics of injected promoters and presence in tested array line (PX819) and integrated lines derived from that array.

| Promoter | Promoter size (bp) | Expected expression location | Array | Integrated |
|---|---|---|---|---|
| *aha-1* | 330 | Neurons, hypodermis, intestine, pharynx (*Jiang et al., 2001*) | Y | Y |
| *hlh-16* | 514 | Head neurons (*Bertrand et al., 2011*) | Y | N |
| *ceh-40* | 965 | Dopaminergic neurons (*Sarov et al., 2012*) | Y | Y |
| *ceh-10* | 1172 | Neurons, seam cells (*Reece-Hoyes et al., 2007*) | Y | Y |
| *ahr-1* | 1387 | ALM and RME neurons (*Huang et al., 2004*) | Y | N |
| *mdl-1* | 2000 | Neurons, body wall, pharynx (*Reece-Hoyes et al., 2007*) | Y | Y |
| *egl-43* | 2001 | Neurons, gonad (*Hwang et al., 2007*) | Y | N |
| *ceh-20* | 2015 | Neurons, seam cells, vulva (*Reece-Hoyes et al., 2007*) | Y | Y |
| *ceh-43* | 2096 | Neurons, anterior hypodermis (*Reece-Hoyes et al., 2007*) | Y | Y |
| *daf-7* | 2524 | Nead neurons, coelemocytes, pharynx (*Klabonski et al., 2016*) | Y | Y |
| *lin-11* | 2857 | Neurons, uterus, vulva, head muscle (*Gupta et al., 2003*) | Y | Y |
| *egl-46* | 4477 | Neurons (*Wu et al., 2001*) | N | N |
| *nhr-67* | 5545 | Neurons, excretory cell, rectal valve cell, vulva (*Fernandes and Sternberg, 2007*) | Y | Y |

Y, yes; N, no.

that technology was to produce randomized genomic edits, and it does not produce precise, library integrations into the genome. Recently, another group (*Mouridi et al., 2022*) built on the utility of heatshock Cas9 and integrated three individual sequences from an array. While these prior multiplexed methods made substantial contributions in improving the efficiency of specific transgenesis, none have yet demonstrated multiplexing beyond tens of unique sequences – orders of magnitude below what would be needed for exploratory transgenesis. TARDIS therefore provides the first true library-based approach for multiplexing transgenesis in *C. elegans*.

## TARDIS as a method for creating barcoded individuals

Genetic barcode libraries have been applied to many high-throughput investigations to reduce sequencing costs and achieve a higher resolution within complex pools of individuals. By focusing the sequencing reads on a small section of the genome, a larger number of individual variants can be identified or experimentally followed. This critical advancement has led to the widespread use of barcoding for evolutionary lineage tracking in microbial systems (*Blundell and Levy, 2014*; *Kasimatis et al., 2021*; *Levy et al., 2015*; *Levy, 2016*; *Nguyen Ba et al., 2019*; *Venkataram et al., 2016*) – uncovering the fitness effects of thousands of individual lineages without requiring large coverage depth of the whole genome. In addition to this application, using barcoded individuals can be used to facilitate any application that involves screening a large pool of diverse individuals within a shared environment. For example, barcodes have been used in microbial studies investigating pharmaceutical efficacy (*Smith et al., 2011*) and barcoded variant screening (*Emanuel et al., 2017*). The TARDIS-based system presented here provides an approximately 1000×-fold increase in barcoding throughput in *C. elegans*, making it a unique resource among multicellular models that allows the large diversity pool and design logic of microbial systems to be adapted to animal models.

While we designed our barcode sequence units for the purpose of barcoding individuals, this approach could also prove useful in future optimization and functional understanding of array-based processes. In particular, the high-sequence diversity but identical physical design of the synthetic barcode library may provide a unique window into extrachromosomal array biology that would be helpful in optimizing sequence units for incorporation into heritable TLAs. For example, an unexpected result of the barcoding experiment was the discovery that a small minority of sequences were overrepresented, or 'jackpotted,' in the TLA relative to their frequency in the injection mix (*Figure 3* and *Figure 3—figure supplement 1*). Our expectation was that arrays would form in an equal molar fashion proportional to the injection mix based on the model that arrays are formed by physical ligation of the injected DNA fragments (*Mello et al., 1991*). Deviations from random array incorporation have been observed before, and a bias for incorporating larger fragments has been proposed as an explanatory mechanism (*Lin et al., 2021*). Our results suggest that the ultimate array composition is not directly proportional to the molarity of the injected fragments or strictly weighted towards the size of the fragment as has been suggested. In contrast, we propose that array size affects the maintenance of extrachromosomal arrays. As such, selection can act to increase the rate of recovery for arrays that have increased their size through random amplification of some sequences by an unknown process early in the formation of the array or by expansion of similar sequences by DNA polymerase slippage during replication, as has been well documented for native chromosomes (*Levinson and Gutman, 1987*). These hypotheses would be consistent with observations of *Lin et al., 2021* if the underlying mechanism for their observation is that inclusion of larger fragments tends to be positively correlated with ultimate array size, and therefore likelihood of maintenance.

## TARDIS as a method for the introduction of promoters and other large constructs

While the barcode approach demonstrates the potential for using TARDIS to integrate large numbers of 433 bp PCR products, previous work using CRISPR/Cas9-initiated homology-directed repair has suggested that integration efficiencies decrease with the size of the insert (*Dickinson and Goldstein, 2016*). We therefore implemented TARDIS for integrating promoters cloned into a vector backbone and ranging in size from 330 bp to 5.5 kb to determine TARDIS functionality under a physically different use case directed specifically at functional analysis. We found that promoter libraries could be integrated into either single sites or two sites simultaneously. Unsurprisingly, the frequency at which various promoters were recovered varied from array to array (e.g., *ahr-1p* was never recovered

in the single-site integration experiment despite being present in the array, while it was the most common promoter recovered in the two-site integration experiment) and likely reflects the same relationship between integration frequency and prevalence in the array, as was seen with the distribution of insert abundance for the barcodes. While we showed that plasmid donors can be used in the TARDIS pipeline, not all arrays contained all 13 plasmids. Given that the estimated 1–13 MB size of arrays (*Carlton et al., 2022*) would be adequate to hold copies of each of the plasmids, as well as the extreme diversity obtained when using smaller DNA fragments, differential presence of a given promoter fragment was somewhat unexpected. This may reflect a preferential use of linear fragments in the in situ assembly of arrays. Future use of linear fragments where feasible may increase incorporation and overall diversity (*Priyadarshini et al., 2022*).

For both the one- and two-site promoter library integrations, transgenic individuals were readily detected, suggesting that the TARDIS method for integration was highly efficient. It has long been understood that successful CRISPR editing at one site significantly increases the chances of successful editing at a second site. This is the premise behind commonly used co-conversion screening strategies (also referred to as co-CRISPR), such as the *dpy-10* screen commonly used in *C. elegans* (*Arribere et al., 2014*; *Ward, 2015*). Here, we show that same type of co-conversion also occurs when using only 'large' (>1 kb), plasmid-based repair templates containing gene-sized repair constructs. Additionally, we have simultaneously targeted the same two landing pads presented here using standard CRISPR techniques and find that approximately half of hygromycin resistant individuals also have rescue of the Unc phenotype (i.e., editing has occurred at both sites; data not shown). Given the high rate of co-conversion, this work demonstrates multiplex integrations are possible not only by targeting multiple repair templates to a single site but also by simultaneously utilizing multiple insertion sites.

In order to recover individual edits most efficiently, given the high frequency of integration using TARDIS, we recommend to either heat-shock small cohorts of array-bearing individuals, such that most cohorts only yield one edited individual or to screen multiple individuals per cohort. Additionally, while split-selection methods allow for direct verification of integration, depending on the downstream use case, integrations should be confirmed by sequencing as errors can still occur, including internal deletions within the insert.

## Expansion of TARDIS to other multicellular systems

Unlocking the investigative potential of transgenesis in animal systems would enable exploratory experiments normally restricted to single-cell models. For example, alanine scanning libraries and protein–protein interactions (*Cunningham and Wells, 1989*; *Matthews, 1996*; *Wells, 1991*), CRISPR library screening (*Bock et al., 2022*), and promoter library generation (*Delvigne et al., 2015*; *Zaslaver et al., 2006*). While we demonstrate the use of TARDIS in *C. elegans* here, the intellectual underpinnings of the approach are agnostic to the research model used. Conceptually, TARDIS facilitates high-throughput transgenesis by using two engineered components: a heritable TARDIS library containing multiplexed transgene units and a genomic split selection landing pad that facilitates integration of single-sequence units from the library. To generate the first TARDIS libraries, we capitalized on the endogenous capacity of *C. elegans* to assemble experimentally provided DNA into heritable extrachromosomal arrays. Extrachromosomal arrays are formed from exogenous DNA, are megabases in size (*Lin et al., 2021*; *Woglar et al., 2020*), do not require specific sequences to form and replicate, and can be maintained in a heritable manner via selection (*Mello et al., 1991*). These qualities make them suitable for use as a heritable library upon which TARDIS can be based. To adopt TLAs in systems beyond *C. elegans*, methods must be adopted to introduce large heritable libraries into the germline as most systems do not maintain extrachromosomal arrays. In mice, the locus *H11* has been used for large transgenic insertions (*Liu et al., 2022*), while in *Drosophila*, the use of PhiC31-mediated transgenesis coupled with bacterial artificial chromosomes (BACs) has allowed for many approximately 10 kb+-sized fragments to be integrated into their respective genomes (*Venken et al., 2006*). Each of these large integration strategies can provide a vehicle for stable inheritance of a TLA.

The second component of the TARDIS integration system is a pre-integrated landing pad sequence. We have generally favored split selectable landing pads (SSLPs) that use HygR for its effectiveness (*Mouridi et al., 2022*; *Stevenson et al., 2020*; *Stevenson et al., 2021*). The SSLPs are engineered to accept experiment-specific units from the array. For example, here we used SSLPs designed to accept barcodes for experimental lineage tracking and promoters for generation of transcriptional reporters.

To translate TARDIS to other systems, a genomic site needs to be engineered to act as a landing pad that can utilize sequence units from the TLA and can be customized to the specific system and use. Because TLAs allow the experimenter to design the library of interest and the landing pad to recapitulate the strengths of single-cell systems, adoption of TARDIS in multicellular animal experiments can leverage the high-resolution, high-diversity exploratory space of DNA synthesis. In addition to adapting assays currently restricted to single-cell models, TARDIS also opens the door to animal-specific uses, such as developmental biology, neurobiology, endocrinology, and cancer research.

In developmental genetics, the lack of large-library transgenesis has resulted in 'barcode' libraries in a different form, utilizing randomized CRISPR-induced mutations to form a unique indel. For example, GESTALT (*McKenna et al., 2016*) creates a diversity of barcodes in vivo via random indel formation at a synthetic target location. LINNAEUS (*Spanjaard et al., 2018*) similarly utilizes randomized targeting of multiple RFP transgenes to create indels, allowing for cells to be barcoded for single-cell sequencing. TARDIS barcodes do not rely on randomized indel generation and thus can be much simpler to implement with sequencing approaches outlined above.

In vivo cancer models have also adopted the high-resolution, high-variant detection of barcodes for the study of tumor growth and evolution. Rogers et al. developed Tuba-seq (*Rogers et al., 2017*; *Winslow, 2022*), a pipeline that takes advantage of small barcodes allowing for in vivo quantification of tumor size. In Tuba-seq, barcodes are introduced via lentiviral infection, leading to the barcoding of individual tumors. TARDIS brings the multiplexed library into the animal context without requiring viral vectors or intermediates, thereby allowing large in vivo library utilization and maintenance. Capitalizing on the large-sequence diversity possible within synthesized DNA libraries with a novel application in multicellular systems generates new opportunities for experimental investigation in animal systems heretofore only possible within microbial models.

## Conclusion

In conclusion, here we have presented TARDIS, a simple yet powerful approach to transgenesis that overcomes the limitations of multicellular systems. TARDIS uses synthesized sequence libraries and inducible extraction and integration of individual sequences from these heritable libraries into engineered genomic sites to increase transgenesis throughput up to 1000-fold. While we demonstrate the utility of TARDIS using *C. elegans*, the process is adaptable to any system where experimentally generated genomic loci landing pads and diverse, heritable DNA elements can be generated.

# Materials and methods

**Key resources table**

| Reagent type (species) or resource | Designation | Source or reference | Identifiers | Additional information |
|---|---|---|---|---|
| Genetic reagent (*Caenorhabditis elegans*) | aha-1p | wormbase.org | WBGene00000095 | |
| Genetic reagent (*C. elegans*) | hlh-16p | wormbase.org | WBGene00001960 | |
| Genetic reagent (*C. elegans*) | ceh-40p | wormbase.org | WBGene00000461 | |
| Genetic reagent (*C. elegans*) | ceh-10p | wormbase.org | WBGene00000435 | |
| Genetic reagent (*C. elegans*) | ahr-1p | wormbase.org | WBGene00000096 | |
| Genetic reagent (*C. elegans*) | mdl-1p | wormbase.org | WBGene00003163 | |
| Genetic reagent (*C. elegans*) | egl-43p | wormbase.org | WBGene00001207 | |

*Continued on next page*

*Continued*

| Reagent type (species) or resource | Designation | Source or reference | Identifiers | Additional information |
|---|---|---|---|---|
| Genetic reagent (*C. elegans*) | *ceh-20p* | wormbase.org | WBGene00000443 | |
| Genetic reagent (*C. elegans*) | *ceh-43p* | wormbase.org | WBGene00000463 | |
| Genetic reagent (*C. elegans*) | *daf-7p* | wormbase.org | WBGene00000903 | |
| Genetic reagent (*C. elegans*) | *lin-11p* | wormbase.org | WBGene00003000 | |
| Genetic reagent (*C. elegans*) | *egl-46p* | wormbase.org | WBGene00001210 | |
| Genetic reagent (*C. elegans*) | *nhr-67p* | wormbase.org | WBGene00003657 | |
| Strain, strain background (*C. elegans*) | N2 | Caenorhabditis Genetics Center | | |
| Strain, strain background (*C. elegans*) | N2-PD1073 | doi:10.17912/micropub.biology.000518 | | Available from the *Caenorhabditis* Intervention Testing Program- upon request (https://citp.squarespace.com/)- |
| Strain, strain background (*C. elegans*) | PX740 | This paper | | N2-PD1073 fxIs47 [*rsp-0p*:: 5′ ΔHygR:: GCGAAGTGACGGTAGACCGT:: 3′ ΔHygR::unc-54 3′::loxP] |
| Strain, strain background (*C. elegans*) | GT331 | This paper | | aSi9[*lox2272 Cbr-unc-119(+) lox2272+loxP 3′3′ ΔHygR +3′ ΔmScarlet-I::PEST*]; *unc-119(ed3)* |
| Strain, strain background (*C. elegans*) | GT332 | This paper | | aSi10[*lox2272 Cbr-unc-119(+) lox2272+loxP 3′ ΔHygR +3′ ΔmScarlet-I*]; *unc-119(ed3)* |
| Strain, strain background (*C. elegans*) | GT336 | This paper | | aSi12[*lox2272 rps-0p::HygR+hsp−16.41p::Cre::tbb-2 3′UTR+sqt-1(e1350) lox2272+loxN 3′ ΔCbr-unc-119(+)::tjp2a_guide:: 3′ ΔmNeonGreen::PEST::egl-13nls::tbb-2 3′UTR*] aSi9[*lox2272 Cbr-unc-119(+) lox2272+loxP 3′ΔHygR::tjp2a guide::3′ΔmScarlet-I::PEST::egl-13nls::tbb-2 3′UTR*] II; *unc-119(ed3)* III |
| Strain, strain background (*C. elegans*) | GT337 | This paper | | aSi13[*lox2272+loxN 3′ ΔCbr-unc-119(+)+3′ ΔmNeonGreen::PEST*] aSi14[*lox2272+loxP 3′ ΔHygR +3′ ΔmScarlet-I::PEST*]; *unc-119(ed3)*, |
| Strain, strain background (*C. elegans*) | QL74 | Gift from QueeLim Ch'ng | | oxEx1578 [*eft-3p::GFP+Cbr-unc-119(+)*] 6x outcross EG4322 |
| Strain, strain background (*C. elegans*) | PX786 | This paper | | fxEx23 [TARDIS #5 *5′ ΔHygR::Intron5'::Read1::NNNCNNTNTNANNNN::Read2::Intron3'::3′ ΔHygR* (89 Unique Sequences) *hsp-16.41p::piOptCas9::tbb-2 34' UTR+rsp-27p::NeoR::unc-54 3′ UTR+U6p::* GCGAAGTGACGGTAGACCGT]; fxSi47[ *rsp-0p:: 5′ ΔHygR::* GCGAAGTGACGGTAGACCGT:: 3′ ΔHygR::unc-54 3′::loxP] |
| Strain, strain background (*C. elegans*) | PX816 | This paper | | fxEx25 [TARDIS #1 *5′ ΔHygR::Intron5'::Read1::NNNCNNTNTNANNNN::Read2::Intron3':: 3′ ΔHygR* (1,319 Unique Sequences) *hsp-16.41p::piOptCas9::tbb-2 34' UTR+rsp-27p::NeoR::unc-54 3′ UTR+U6p::* GCGAAGTGACGGTAGACCGT]; fxSi47[ *rsp-0p:: 5′ ΔHygR::* GCGAAGTGACGGTAGACCGT:: 3′ ΔHygR::unc-54 3′::loxP] |

*Continued on next page*

*Continued*

| Reagent type (species) or resource | Designation | Source or reference | Identifiers | Additional information |
|---|---|---|---|---|
| Strain, strain background (*C. elegans*) | PX817 | This paper | | fxEx26 [TARDIS #2 5' *ΔHygR::Intron5'::Read1::NNNCNNTNTNANNNN::Read2::Intron3'::* 3' *ΔHygR* (3,001 Unique Sequences) *hsp-16.41p::piOptCas9::tbb-2 34' UTR+rsp-27p::NeoR::unc-54 3' UTR+U6p::* GCGAAGTGACGGTAGA CCGT]; fxSi47[ *rsp-0p::* 5' *ΔHygR::* GCGAAGTGACGGTAGACCGT:: 3' *ΔHygR::unc-54 3'::loxP]* |
| Strain, strain background (*C. elegans*) | PX818 profile 1 | This paper | | fxEx27 [TARDIS #3 5' *ΔHygR::Intron5'::Read1::NNNCNNTNTNANNNN::Read2::Intron3'::* 3' *ΔHygR* (91 Unique Sequences) *hsp-16.41p::piOptCas9::tbb-2 34' UTR+rsp-27p::NeoR::unc-54 3' UTR+U6p::* GCGAAGTGACGGTAGA CCGT]; fxSi47[ *rsp-0p::* 5' *ΔHygR::* GCGAAGTGACGGTAGACCGT:: 3' *ΔHygR::unc-54 3'::loxP]* |
| Strain, strain background (*C. elegans*) | PX818 profile 2 | This paper | | fxEx28 [TARDIS #4 5' *ΔHygR::Intron5'::Read1::NNNCNNTNTNANNNN::Read2::Intron3'::* 3' *ΔHygR* (204 Unique Sequences) *hsp-16.41p::piOptCas9::tbb-2 34' UTR+rsp-27p::NeoR::unc-54 3' UTR+U6p::* GCGAAGTGACGGTAGA CCGT]; fxSi47[ *rsp-0p::* 5' *ΔHygR::* GCGAAGTGACGGTAGACCGT:: 3' *ΔHygR'::unc-54 3'::loxP]* |
| Strain, strain background (*C. elegans*) | PX819 | This paper | | N2 fxEx24 [(*rps-0p::* 5' *ΔHygR+loxP + aha-1p::SV40 NLS::* 5' *ΔmScarlet-I*) + (*rps-0p::* 5' *ΔHygR+loxP + ahr-1p::SV40 NLS::*5' *ΔmScarlet-I*) + (*rps-0p::* 5' *ΔHygR+loxP + ceh-10-1p::SV40 NLS::*5' *ΔmScarlet-I*) + (*rps-0p::* 5' *ΔHygR+loxP + ceh-20p::SV40 NLS::*5' *ΔmScarlet-I*) + (*rps-0p::* 5' *ΔHygR+loxP + ceh-40p::SV40 NLS::*5' *ΔmScarlet-I*) + (*rps-0p:: ΔHygR+loxP + ceh-43p::SV40 NLS::*5' *ΔmScarlet-I*) + (*rps-0p::* 5' *ΔHygR+loxP + daf-7p::SV40 NLS::*5' *ΔmScarlet-I*) + (*rps-0p:: ΔHygR+loxP + egl-43p::SV40 NLS::*5' *ΔmScarlet-I*) + (*rps-0p::* 5' *ΔHygR+loxP + hlh-16p::SV40 NLS::*5' *ΔmScarlet-I*) + (*rps-0p::* 5' *ΔHygR+loxP + lin-11p::SV40 NLS::*5' *ΔmScarlet-I*) + (*rps-0p::* 5' *ΔHygR+loxP + mdl-1p::SV40 NLS::*5' *ΔmScarlet-I*) + (*rps-0p::* 5' *ΔHygR+loxP + nhr-67p::SV40 NLS::*5' *ΔmScarlet-I*)+*hsp−16.41p::piOptCas9::tbb-2 34' UTR+prsp-27::NeoR::unc-54 3' UTR];* aSi10[lox2272+*Cbr-unc-119*(+)+lox2272+loxP + 5' *ΔHygR::unc-54 3' UTR*+5' *ΔmScarlet-I::egl-13 NLS::tbb-2 3' UTR*, II:8420157]; *unc-119(ed3)* III |
| Strain, strain background (*C. elegans*) | EG4322 | doi.org/10.1038ng. 248; Caenorhabditis Genetics Center | | |
| Strain, strain background (*Escherichia coli*) | PXKR1 | This paper | | NA22 transformed with pUC19 |
| Recombinant DNA reagent | Plasmid pDSP15 | This paper | 193853 (Addgene) | 5' *ΔHygR::loxP::MCS::5' Δ mScarlet-I* |
| Recombinant DNA reagent | Plasmid pDSP16 | This paper | 193854 (Addgene) | 5' *ΔCbr-unc-119(+)::loxN::MCS::5' Δ 5'mNeonGreen* |
| Recombinant DNA reagent | Plasmid pMS84 | This paper | 193852 (Addgene) | *U6p::*GGACAGTCCTGCCGAGGTGG |
| Recombinant DNA reagent | Plasmid pZCS36 | This paper | 193048 (Addgene) | *hsp16.41p::Cas9(dpiRNA)::tbb-2 '3UTR* |
| Recombinant DNA reagent | Plasmid pZCS38 | This paper | 193049 (Addgene) | *rsp-27p::NeoR::unc-54 3' UTR* |
| Recombinant DNA reagent | Plasmid pZCS41 | This paper | 193050 (Addgene) | *U6p::*GCGAAGTGACGGTAGACCGT |
| Sequence-based reagent | ZCS422 | This paper | | Design and construction of barcode donor library |

*Continued on next page*

*Continued*

| Reagent type (species) or resource | Designation | Source or reference | Identifiers | Additional information |
|---|---|---|---|---|
| Commercial assay or kit | DNA Clean and Concentrator | Zymo Research | Cat# D4004 | |
| Commercial assay or kit | Genomic DNA Clean and Concentrator | Zymo Research | Cat# D4011 | |
| Commercial assay or kit | Zymoclean Gel DNA Recovery Kit | Zymo Research | Cat# D4008 | |
| Commercial assay or kit | Zyppy Plasmid Miniprep Kit | Zymo Research | Cat# D4019 | |
| Software, algorithm | Cutadept | doi.org/10.14806/ej.17.1.200 | Version 4.1 | |
| Software, algorithm | AmpUMI | doi.org/10.1093/bioinformatics/bty264 | Version 1.2 | |
| Software, algorithm | Starcode | doi.org/10.1093/bioinformatics/btv053 | Version 1.4 | |
| Software, algorithm | Google colab | colab.research.google.com | | |
| Software, algorithm | Python (version) | *Guido van Rossum, 1991* | Version 3.7.13 | |
| Software, algorithm | Juypter Notebook (IPython) | doi:10.3233/978-1-61499-649-1-87 | Version 7.9.0 | |
| Software, algorithm | matplotlib | doi:10.5281/zenodo.3898017 | Version 3.7.13 | |
| Software, algorithm | Fiji | imagej.net/software/fiji/ | Version 2.9.011.53t | |
| Chemical compound, drug | G-418 | GoldBio (CAS number 108321-42-2) | Cat# G-418-5 | |
| Chemical compound, drug | Hygromycin B | GoldBio (CAS number 31282-04-9) | Cat# H-270-10-1 | |

## General TARDIS reagents

Strains generated for this publication along with key plasmids and reagents are listed in the Key Resources Table. A full list of all plasmids is given in *Supplementary file 1*. All plasmids were cloned by Gibson Assembly following the standard NEB Builder HiFi DNA Assembly master mix protocol (New England Bio Labs [NEB], MA), unless otherwise indicated. All plasmids have been confirmed by restriction digest, Sanger sequencing, and/or full plasmid sequencing. All primers used in the construction and validation of plasmids are listed in *Supplementary file 2*.

To generate our heatshock-inducible Cas9, *hsp16.41p::Cas9dpiRNA::tbb-2 '3UTR,* the *hsp16.41* promoter was amplified from pMA122 (Addgene ID34873) (*Frøkjær-Jensen et al., 2012*). The germline-licensed Cas9 and tbb-2 3' UTR were amplified from pCFJ150-Cas9 (dpiRNA) (Addgene ID107940) (*Zhang et al., 2018*). All fragments were assembled into PCR-linearized pUC19 vector (NEB) to give the final plasmid pZCS36.

To generate a standard empty guide vector, *U6p*::(empty)gRNA, the U6p and gRNA scaffold from pDD162 (Addgene ID47549) (*Dickinson et al., 2015*) was amplified and assembled into PCR-linearized pUC19 to generate pZCS11.

To generate *rsp-27p::NeoR::unc-54 3' UTR,* the full resistance cassette was amplified from pCFJ910 (Addgene ID44481) and assembled into PCR-linearized pUC19 vector to give pZCS38.

## Genomic DNA isolation for array and integrant characterization

For processing large populations of worms, a widely used bulk lysis protocol was adapted (Fire Lab 1997 Vector Supplement, February 1997). In brief, 450 µl of worm lysis buffer (0.1 M Tris-Cl pH 8.0, 0.1 M NaCl, 50 mM EDTA pH 8.0, and 1% SDS) and 20 µl 20 mg/ml proteinase K were added to approximately 50 µl of concentrated worm pellet. Samples were inverted several times to mix and incubated at 62°C for 2 hr. After incubation, samples were checked under the microscope to ensure no visible worm bodies were left in the solution. ChIP DNA binding buffer (Zymo, CA) was added in a 2:1 ratio and gently inverted to mix. Samples were then purified with Zymo-Spin IIC-XLR columns following the manufacturer's protocol. Samples were eluted in 50 µl of water. Each sample was then digested with 10 mg/ml RNase A (Thermo Fisher Scientific, MA, Cat# EN0531) at 42°C for 2 hr. Genomic DNA was then reisolated by adding a 2:1 ratio of ChIP DNA binding buffer and purifying with Zymo-Spin IIC-XLR columns. Final genomic samples were quantified by Nanodrop.

For individual worm lysis, individual array-bearing worms were isolated and lysed in 4 µl of EB (Zymo, Cat# D3004-4-16) buffer with 1 mg/ml proteinase K (NEB). Each sample was rapidly frozen in liquid nitrogen and then thawed to disrupt the cuticle and then incubated at 58°C for 1 hr, with a subsequent incubation at 95°C for 20 min to inactivate the proteinase K.

## TARDIS integration: General protocol

On day 0, TARDIS array-bearing *C. elegans* grown to a high density of gravid adults were hypochlorite synchronized in NGM buffer (*Leung et al., 2011*) and grown overnight in 15 ml NGM with G-418 (1.56 mg/ml) at 15°C with nutation. On day 1, L1s were washed three times with NGM buffer to remove G-418, plated onto media without selective agent, and continued to be grown at 15°C. On day 2, L2/L3s were heat-shocked at 35.5°C for 1 hr. After heat shock, worms were grown at 25°C until gravid adults when hygromycin B was top spread on plates at a final concentration of 250 µg/ml.

## Construction of landing pad for barcodes

To create the barcode landing pad, an intermediate Chr. II insertion vector, pZCS30, was built from pMS4 by using PCR to remove the *let-858* terminator. pZCS30 served as the vector backbone for pZCS32. To assist in cloning, the backbone was split into two PCR fragments. The broken *HygR* gene was amplified in two parts, *rsp-0p::5'ΔHygR* and *3'ΔHygR::unc-54 3' UTR*, from pCFJ1663 (Addgene ID51484). Overlapping PCR was used to fuse both *HygR* fragments. The resulting broken *HygR* cassette removed the intron found in pCFJ1663 as well as four codons from exon 1 and three codons from exon 2, while also creating +1 frameshift and a reverse orientation guide RNA target for pZCS41. A second overlapping PCR was performed to fuse the broken *HygR* cassette to backbone fragment 2. The resulting two-part clone was then assembled to give pZCS32.

The barcode landing pad TARDIS strain, PX740, was created by injecting a mixture of 10 ng/µl pZCS32, 50 ng/µl pMS8, and 3 ng/µl pZCS16 (Addgene ID154824) (*Stevenson et al., 2020*) into the gonad of young adult N2-PD1073 (*Teterina et al., 2022*) hermaphrodites. Screening and removal of the SEC were performed following *Dickinson et al., 2015*. Presence of the correct insertion was confirmed by Sanger sequencing using the primers listed in *Supplementary file 3*.

To create the barcode landing pad targeting guide RNA, *U6p*:: GCGAAGTGACGGTAGACCGT, the guide sequence GCGAAGTGACGGTAGACCGT was added by overlapping primers to the vector pZCS11 to give the final construct pZCS41.

## Design and construction of barcode donor library

Oligo ZCS422 was ordered with 11 randomized N's (hand-mixed bases) (Integrated DNA Technologies [IDT], IA) and has the following sequence: CTACACGACGCTCTTCCGATCTNNNCNNTNTNANNNNA GATCG GAAGAGCACACGTCTG. Four 'hard-coded' base pairs were included within the randomized sequence. ZCS422 was used as the core for the generation of two separate complexing PCR barcode homologies referred to as 'barcode-15X' and 'barcode-20X' to denote the number of complexing cycles (*Figure 2*). All PCRs were performed using the high-fidelity Q5 polymerase (NEB) as per the manufacturer's instructions. All primers used for barcode synthesis can be found in *Supplementary file 4*. For both 'barcode-15X' and 'barcode-20X,' the left and right homology arms were generated separately by PCR and purified by gel extraction. An initial 10-cycle PCR was performed to convert the oligo into a 201 bp double-stranded product that was gel extracted with Zymo clean Gel DNA

Recovery Kit (Cat# D4008) following the manufacturer's protocol. The low cycle number was done to retain diversity and minimize the effects of PCR jackpotting.

For 'barcode-15X,' to generate the complete donor homology, the double-stranded barcode template was combined with both the left and right homology arms for a three-fragment overlap PCR. To maximize diversity, high concentrations of the individual templates were used. The reaction contained 52 fmol/µl of barcode template and 22 fmol/µl of left right arms in a single 50 µl Q5 reaction. A total of 15 cycles were performed. The lower cycle was again done to reduce PCR jackpotting. The single product was gel extracted as a 433 bp fragment. The final donor fragment is referred to as 'barcode-15X.'

To generate 'barcode-20X,' a similar three-fragment overlap PCR was used. 4.3 fmol/µl of barcode template, 15.33 fmol/µl of left arm, and 3.3 fmols/µl of right arm were combined across six Q5 50 ul reactions and a total of 20 cycles were performed. The right arm concentration was lower caused by low concentration extraction. The single product was gel extracted as a 433 bp fragment. The final donor fragment is referred to as 'barcode-20X.'

## Generation of barcode TLA lines

The TARDIS array-bearing line PX786 was created by injecting 50 ng/µl of barcode-15X, 10 ng/µl pZCS38, 15 ng/µl pZCS41, 5 ng/µl pZCS16, and 20 ng/µl pZCS36 into young adult PX740 hermaphrodites. Individual injected worms were grown at 15°C for 4 d and then treated with G-418 (1.56 mg/ml). A single stable array line was isolated and designated PX786.

The TARDIS array-bearing lines PX816, PX817, PX818 profile 1 and PX818 profile 2 were created by injecting 100 ng/µl of barcode-20X, 10 ng/µl pZCS38, 15 ng/µl pZCS41, and 20 ng/µl pZCS36. Individual injections were grown at 15°C for 4 d and then treated with G-418 (1.56 mg/ml). Full genotypes are provided in *Supplementary file 7* as the full genotypes cannot be contained within a table.

## Estimation of barcode integration frequency population sample preparation

PX786 was grown to gravid adults in the presence of G-418 with concentrated NA22 transformed with pUC19 for ampicillin resistance as a food source (designated PXKR1). Once gravid, the strain was hypochlorite synchronized and grown overnight in 15 ml NGM buffer with G-418 at 15°C with nutation. For each of the four replicates, a synchronized L1 population was divided in half. The first half was pelleted by centrifugation (2400 × *g* for 2 min) and frozen (–20°C) until processed. These samples represented the array-bearing samples. Another sample of approximately 150,000 L1s was plated to large NGM and subjected to the standard TARDIS heat shock and grown until the population was primarily gravid adults. Then, this population was hypochlorite synchronized and grown in NGM buffer at 15°C with hygromycin B (250 µg/ml). These entire samples were pelleted and frozen, representing the F1 samples.

## PCR for barcode quantification

Several different PCRs were performed depending on the intended downstream sequencing quantification. See *Figure 2—figure supplement 1* for a schematic layout of the different PCR steps. The primers used for barcode quantification are given in *Supplementary file 5*. To quantify the diversity of arrays from either a bulk population or individual worms, two separate PCRs were performed to quantify the diversity of arrays.

The first PCR (Amplicon one array) was performed for three cycles to add Unique Molecular Identifiers (UMI), allowing for downstream de-duplication. For each sample, either 100 ng of genomic DNA (bulk samples) or the entirety of the worm lysate (single worms) was used as the template. PCR samples were then purified using the Zymo DNA Clean and Concentrator-5 Kit (Cat# D4004) following the manufacturer's protocol and eluted with 24 µl water. Samples were not quantified prior to the next step as most DNA was not from the target PCR product. A second PCR (Amplicon two) using the entire 24 µl of the extract from the previous step was performed for 24 cycles to add indices. In some cases, a smaller, nonspecific product was also formed, so each sample was run on a 2% agarose gel and extracted for the 169 bp size product.

Two separate PCRs were performed to quantify the diversity of integrated barcode sequences. The first PCR (Amplicon one integrant) was performed for three cycles to add UMI sequences. For

each sample, 100 ng of genomic DNA was used as the template. PCR products were then purified as described above and followed the Amplicon two protocol. Each product was quantified on a Synergy H1 plate reader using software Gen 5 3.11. Samples were mixed at an equal molar ratio for a 20 nM final concentration for Illumina sequencing.

## Illumina sequencing and data processing for barcode characterization

To quantify the diversity of barcodes in each sample, PCR products were sequenced on either a single NextSeq 500 lane or NovaSeq SP, with single read protocols performed by the Genomics and Cell Characterization Facility (GC3F) at the University of Oregon. Compressed fastq files were processed with cutadept 4.1 (*Martin, 2011*) to remove low-quality reads (quality score < 30, max expected error = 1, presence of 'N' within the read) and trimmed to 87 bp. For the NextSeq lane, the specific nextseq trim = 30 command was used. The sequences were then demultiplexed using cutadept. For duplicate removal, AmpUMI (*Clement et al., 2018*) in 'processing mode' was used with umi regex 'CACIIIIIIIIIGAC' for individual index files. De-duplicated reads were then trimmed to 15 base pairs with cutadapt for each file. Starcode (*Zorita et al., 2015*) was then used for mutation correction and counting of each barcode sequence. Each unique sequence was only kept if its final length was 15 base pairs. For the injection mix, each unique barcode was kept regardless of total reads. For all TARDIS arrays and F1 integrations, we used the observed plateau in the number of observed unique barcodes for various count cutoffs to establish a conservative threshold of five or more reads for true barcode sequence (*Figure 3—figure supplement 3*). Visualizations were created with Python 3.7.13 (*Guido van Rossum, 1991*) and matplotlib 3.5.2 (*Hunter, 2007*). Sequence logos were created with Logomaker (*Tareen and Kinney, 2020*). Correlation and p-values were generated by scipy input stats. pearsonr (*Virtanen et al., 2020*). This statistical test was chosen because the relationship from array to integration is approximately linear. All data were processed in Jupyter Notebooks (*Kluyver et al., 2016*) utilizing Google Colaboratory (colab.research.google.com). All Python code is available on Figshare.

## Design of landing pads for transcriptional reporters

The utilized fluorophores, mScarlet-I (*Bindels et al., 2017*) and mNeonGreen (*Shaner et al., 2013*), were synthesized with the desired modifications as genes incorporated into the pUCIDT-KAN plasmid (IDT). First, a SV40 nuclear localization sequence (NLS) was added after the 13th codon of the *mScarlet-I* gene. This same 66 bp sequence was also used in place of the first four codons of the *mNeonGreen* gene. Secondly, a PEST domain (*Li et al., 1998*) flanked by MluI restriction endonuclease sites and an additional NLS from the *egl-13* gene (*Lyssenko et al., 2007*) were added to the 3' end of the genes. The *C. elegans* Codon Adapter (https://worm.mpi-cbg.de/codons/cgi-bin/optimize. py; *Redemann et al., 2011*) was used to codon optimize both modified fluorophore sequences and identify locations for three synthetic introns. The first two introns contained 10-base pair periodic $A_n/T_n$-clusters (PATCs), which have been shown to reduce the rates of transgene silencing (*Frøkjær-Jensen et al., 2016*), while the third was a standard synthetic intron. Finally, the 3' UTR of the *tbb-2* gene, which is permissive for germline expression (*Merritt et al., 2008*), was added to the end of fluorophore genes. The modified *mScarlet-I* and *mNeonGreen* genes were PCR amplified and assembled into NotI and SnaBI-linearized pDSP1, a standard backbone vector derived from pUCIDT-KAN. The resulting *mScarlet-I*-containing plasmid was designated pDSP6 and the *mNeonGreen*-containing plasmid was designated pDSP7. In addition, pDSP9, a version of mScarlet-I lacking the PEST destabilization sequence, was generated by PCR amplifying the shared *egl-13* NLS and *tbb-2* 3' UTR sequence from pDSP6 and then assembling this fragment into MluI and SnaBI-linearized pDSP6.

Landing pads were built using a modification of our previous split landing pad strategy (*Stevenson et al., 2020*). Each landing pad contained the 3' portion of a selectable marker followed by a validated guide sequence and the 3' portion of a fluorophore. The guide sequence (GGACAGTCCTGCCGAG GTGGAGG) has no homology in the *C. elegans* genome and has been previously shown to allow for efficient editing (*Stevenson et al., 2020*). This sequence was targeted by the plasmid pMS84, which was made from pZCS2, a plasmid made in the same manner as pZCS11 but which is missing a segment of the plasmid backbone, using the Q5 site-directed mutagenesis kit (NEB). *mScarlet-I* was paired with a *HygR* marker (*Dickinson et al., 2013*) while *mNeonGreen* gene was paired with the *Cbr-unc-119(+)* rescue cassette (*Frøkjaer-Jensen et al., 2008*).

## Construction of split *HygR/mScarlet-I* landing pads

The split *HygR/mScarlet-I* landing pad was inserted into the well-characterized *ttTi5605* Mos1 site on Chromosome II (*Frøkjaer-Jensen et al., 2008*). pQL222 (a gift from Dr. QueeLim Ch'ng), a modified version of the pCFJ350 (*Frøkjær-Jensen et al., 2012*) in which the original resistance marker was changed to a kanamycin and zeocin cassette, was digested with BsrGI to provide a linear vector backbone. The *Cbr-unc-119* gene, with a lox2272 sequence added to the 5′ end, and a multiple cloning site (MCS) with a lox2272 site added to the 3′ end were PCR amplified from pQL222. These two fragments were assembled into the linearized backbone to yield pDSP2.

Next, the 3′ 949 bases of the *HygR* marker were amplified along with the *unc-54* 3′UTR from pDD282 (*Dickinson et al., 2015*). The primers used were designed to invert the loxP sequence at the 3′ end of *unc-54* 3′UTR from its original orientation in pDD282 and to add the guide sequence to the 5′ end of the *HygR* fragment. The 3′ 821 bases of the *mScarlet-I* gene along with the tbb-2 3′ UTR were amplified from pDSP6. These two amplicons were assembled into a SbfI/SnaBI digested pDSP2 vector to yield pDSP61. Similarly, the *mScarlet-I* gene was amplified from pDSP9 and assembled into pDSP2 along with the *HygR* fragment to give pDSP62, a PEST-less version of the landing pad construct. Both the PEST-containing and PEST-less versions of the split *HygR/mScarlet-I* landing pads were integrated into QL74, a 6× outcross of EG4322 (*Frøkjaer-Jensen et al., 2008*), using the standard MosSCI technique (*Frøkjær-Jensen et al., 2012*) to yield strains GT331 and GT332.

## Construction of Split *Cbr-unc-119*(+)/*mNeonGreen* landing pad

To construct the *Cbr-unc-119(+)/mNeonGreen* landing pad, we wanted to find a genomic safe harbor site permissive to germline expression of transgenes. The *oxTi179* universal MosSCI site on Chromosome II permits germline expression but interrupts *arrd-5*, an endogenous *C. elegans* gene. Therefore, CRISPR-mediated genome editing was used to place the landing pad between *ZK938.12* and *ZK938.3*, two genes adjacent to *arrd-5* whose 3′ UTRs face each other. The genomic sequence catg gtataaagtgaatca<u>AGG</u> was targeted by the plasmid pDSP45 which was made from pDD162 (*Dickinson et al., 2013*) using the Q5 site-directed mutagenesis kit (NEB).

Chromosomal regions II:9830799–9831548 and II:9831573–9832322 were amplified from genomic DNA for use as homology arms. The self-excising cassette (SEC) was PCR amplified from pDD282 such that the loxP sites were replaced by lox2272 sites. An MCS was amplified from pDSP2 while a linear vector backbone fragment was amplified from pDSP1. All five of these PCR fragments were assembled into a circular plasmid, which was immediately used as a template for seven synonymous single-nucleotide substitutions into the terminal 21 bp of the *ZK938.12* gene fragment by Q5 site-directed mutagenesis kit (NEB). The resultant plasmid was named pDSP47.

The 3′ 846 bases of the *Cbr-unc-119(+)* rescue cassette plus the 3′ UTR were amplified from pDSP2 such that the lox2272 sequence after the 3′ UTR was replaced with a loxN site and the guide site GGACAGTCCTGCCGAGGTGGAGG was added upstream of the coding sequence. The 3′ 818 bases of *mNeonGreen* plus the tbb-2 3′ UTR were amplified from pDSP7. These two amplicons were assembled into StuI/AvrII digested pDSP47 to yield pDSP63.

Following the protocol from *Dickinson et al., 2015*, the landing pad from pDSP63 was integrated into the GT331 strain using pDSP45 as the guide plasmid. Upon integration, this yielded strain GT336. Activation of the Cre recombinase within the SEC by heat shock caused both the removal of the SEC from the *mNeonGreen* landing pad and the *Cbr-unc-119(+)* cassette from the *mScarlet-I* landing pad. The combined effect of this double excision event was to yield strain GT337, which has an Unc phenotype and no longer has the hygromycin resistance and Rol phenotypes.

## Design and construction of promoter library

Targeting vectors were constructed to provide the 5′ portions of each split gene pairing. Both targeting vectors had the same multiple cloning site, allowing promoter amplicons to be assembled into either vector using the same set of primers. In addition, each selectable marker gene is flanked by a lox site that matches the sequence and orientation of the lox site flanking the 3′ portion of the marker in the genomic landing pad, allowing for the optional post-integration removal of the selectable marker gene using Cre recombinase.

To construct the split *HygR/mScarlet-I* targeting vector, the *rps-0* promoter plus the 5′ 627 bases of the *HygR* gene were amplified from pDD282 such that a loxP site was added in front of the promoter

sequence. The MCS was amplified from pDSP2 and the 5′ 803 bases of the *mScarlet-I* gene were amplified from pDSP6. All three of these amplicons were assembled into NotI/SnaBI digested pDSP1 to yield pDSP15.

To construct the split *Cbr-unc-19(+)/mNeonGreen* targeting vector, the promoter and the 5′ 515 bases of the *Cbr-unc-19(+)* were amplified from pDSP2 such that a loxN site was added prior to the promoter. The MCS was amplified from pDSP2 and the 5′ 830 bases of the *mNeonGreen* gene were amplified from pDSP7. All three of these amplicons were assembled into NotI/SnaBI digested pDSP1 to yield pDSP16.

The entire intergenic region was used for *aha-1p*, *ahr-1p*, *ceh-20p*, *ceh-40p*, *egl-46p*, *hlh-16p*, and *nhr-67p*. For *ceh-43p*, the 2096 bp upstream of the *ceh-43* start codon was used. For *mdl-1p*, *egl-43p*, and *ceh-10p*, the promoters describe in *Reece-Hoyes et al., 2013* were used. For *daf-7p* and *lin-11p*, the promoters described in *Entchev et al., 2015* and *Marri and Gupta, 2009*, respectively, were used. Promoters were amplified from N2 genomic DNA using primers designed to add the appropriate homology to the targeting vector and assembled into PCR-linearized pDSP15 or pDSP16 for split *HygR/mScarlet-I* and split *Cbr-unc-19(+)/mNeonGreen*, respectively.

## Insertion of promoter libraries by TARDIS

For integration of a promoter library into a single landing pad site, a mixture consisting of 15 ng/µl guide plasmid (pMS84), 20 ng/µl *hsp16.41::Cas9* plasmid (pZCS36), 10 ng/µl neomycin resistance plasmid (pZCS38), and 0.45 fmol/µl of each of the 13 repair template plasmids (*Table 1*) was micro-injected into the gonad arms of young adult GT332 hermaphrodites. Individuals were incubated at 20°C and after 3 d treated with 1.56 mg/ml G-418 to select for array-bearing individuals. Once stable array lines were obtained, integration was done using the standard TARDIS protocol using a density of approximately 200 L1s per plate.

For integration of a promoter library into two landing pad site, a mixture consisting of 15 ng/µl guide plasmid (pMS84), 20 ng/µl *hsp16.41::Cas9* plasmid (pZCS36), 0.5 ng/µl neomycin resistance plasmid (pZCS38), and 0.45 fmol/µl of each of the 14 repair template plasmids (seven targeted to each site) was microinjected into the gonad arms of young adult GT337 hermaphrodites. Individuals were incubated at 20°C and after 3 d treated with 1.56 mg/ml G-418 to select for array-bearing individuals. Once a stable array line was obtained, plates of mixed stage worms were transferred to plates without drug, heat-shocked at 35.5°C for 1.5 hr and returned to 20°C. Three days after heat shock, hygromycin was added at a final concentration of 250 µg/ml.

For both scenarios, candidate worms (those which had both hygromycin resistance and wild-type movement) were singled and screened by PCR. The identity of the integrated promoters was determined by Sanger sequencing of the PCR product. The primers used for genotyping are given in *Supplementary file 6*.

## Microscopy

Individual late L4/young adults were mounted on 2% agarose pads and immobilized with 0.5 M levamisole. Imaging was performed on a DeltaVision Ultra microscope (Cytiva, MA) using the 20x objective and Acquire Ultra software version 1.2.1. Fluorescent images were acquired using the orange (542/32 nm) and green (525/48 nm) filter sets for mScarlet-I and mNeonGreen, respectively. Light images were captured at 5% transmission and a 0.01 s exposure. Fluorescent images were captured at 5% transmission and a 2 s (*aha-1p*), 1 s (*ceh-40p*, *ceh-43p*, *nhr-67p*, *ceh-10p::mNeonGreen*), 0.5 s (*ceh-10p::mScarlet-I*, *ceh-20p*, *daf-7p*), or 0.2 s (*lin-11p*, *mdl-1p*) exposure. Images were processed in Fiji (ImageJ) version 2.9.0/1.53t.

## Accessibility of reagents, data, code, and protocols

The authors affirm that all data necessary for confirming the conclusions of the article are present within the article, figures, and tables. Plasmids pDSP15 (Addgene ID 193853), pDSP16 (Addgene ID193854), pMS84 (Addgene ID 193852), pZCS36 (Addgene ID 193048), pZCS38 (Addgene ID193049), and pZCS41 (Addgene ID 193050) are available through Addgene and can be freely viewed and edited in ApE (*Davis and Jorgensen, 2022*) and other compatible programs. Strains PX740, GT332, and GT337 are available from the Caenorhabditis Genetics Center (cgc.umn. edu). Strains and plasmids not available at a public repository are available upon request. Illumina

sequencing data are available at BioProject ID: PRJNA893002. All other data, code, plasmid and landing sequences, and original microscopy images are available on Figshare (*Stevenson et al., 2022*). We plan to continue to develop TARDIS technology and provided descriptions of updated libraries and advancements at https://github.com/phillips-lab/TARDIS, (copy archived at *ZCST, 2022*).

## Acknowledgements

We thank the Phillips lab members for helpful suggestions and technical assistance. In particular, we thank Erin Jahahn for her assistance in early TARDIS multiple integrations, Zach Muñoz for his assistance injecting several TARDIS libraries, Kristin Robinson for creating PXKR1, and Ellie Laufer for her assistance in genotyping individual promoter integrations. We also thank Sihoon Moon, Hyun Jee Lee, and Eric Andersen for technical assistance in plasmid construction.

## Additional information

### Competing interests

Zachary C Stevenson, Stephen A Banse, Patrick C Phillips: The presented technology underlies U.S. patent application 17/236,556 and associated U.S. Provisional Application No. 63/013,365 (Inventors ZCS, SAB, PCP, assignee University of Oregon). The patent applicant/assignee and grant funding institutions had no involvement in the described work, including but not limited to experimental design, data analysis, interpretation, or manuscript preparation. The other authors declare that no competing interests exist.

### Funding

| Funder | Grant reference number | Author |
|---|---|---|
| National Institutes of Health | R35GM131838 | Patrick C Phillips |
| National Institutes of Health | R01AG056436 | Hang Lu Patrick C Phillips |

The funders had no role in study design, data collection and interpretation, or the decision to submit the work for publication.

### Author contributions

Zachary C Stevenson, Conceptualization, Resources, Data curation, Software, Formal analysis, Validation, Investigation, Visualization, Methodology, Writing – original draft, Project administration, Writing – review and editing; Megan J Moerdyk-Schauwecker, Conceptualization, Resources, Data curation, Formal analysis, Validation, Investigation, Visualization, Methodology, Writing – original draft, Writing – review and editing; Stephen A Banse, Conceptualization, Formal analysis, Supervision, Visualization, Methodology, Writing – original draft, Project administration, Writing – review and editing; Dhaval S Patel, Resources, Writing – review and editing; Hang Lu, Supervision, Funding acquisition, Project administration, Writing – review and editing; Patrick C Phillips, Conceptualization, Supervision, Funding acquisition, Methodology, Project administration, Writing – review and editing

### Author ORCIDs

Zachary C Stevenson (ID) http://orcid.org/0000-0002-6564-6967
Stephen A Banse (ID) http://orcid.org/0000-0002-5540-4526
Hang Lu (ID) http://orcid.org/0000-0002-6881-660X
Patrick C Phillips (ID) http://orcid.org/0000-0001-7271-342X

Reviewer #1 (Public Review): https://doi.org/10.7554/eLife.84831.3.sa1
Reviewer #2 (Public Review): https://doi.org/10.7554/eLife.84831.3.sa2
Author Response: https://doi.org/10.7554/eLife.84831.3.sa3

## Additional files

### Supplementary files
• Supplementary file 1. Table of the plasmids used.
• Supplementary file 2. Table of the primers used for cloning.
• Supplementary file 3. Table of the primers used for strain confirmation by Sanger sequencing.
• Supplementary file 4. Table of the primers and ultramer used to create the barcode donor homology.
• Supplementary file 5. Table of the primers used for Illumina sequencing.
• Supplementary file 6. Table of the primers used for promoter identification.
• Supplementary file 7. Complete genotypes for the barcode TARDIS arrays presented in the study.
• MDAR checklist

### Data availability

The authors affirm that all data necessary for confirming the conclusions of the article are present within the article,figures, and tables. Plasmids pDSP15 (Addgene ID 193853), pDSP16 (Addgene ID19384), pMS84 (Addgene ID193852), pZCS36 (Addgene ID 193048), pZCS38 (Addgene ID193049), and pZCS41 (Addgene ID 193050), are available through Addgene and can be freely viewed and edited in ApE (Davis and Jorgensen, 2022) and other compatible programs. Strains PX740, GT332 and GT337 are available from the Caenorhabditis Genetics Center (cgc.umn.edu). Strains and plasmids not available at a public repository are available upon request. Illumina sequencing data are available at BioProject ID: PRJNA893002. All other data, code, plasmid and landing sequences and original microscopy images are available on Figshare (*Stevenson et al., 2022*). We plan to continue to develop TARDIS technology and provided descriptions of updated libraries and advancements at: https://github.com/phillips-lab/TARDIS (copy archived at *ZCST, 2022*).

The following dataset was generated:

| Author(s) | Year | Dataset title | Dataset URL | Database and Identifier |
|---|---|---|---|---|
| Stevenson ZC, Moerdyk-Schauwecker MJ, Banse SA, Patel DS, Lu H, Phillips PC | 2022 | High-Throughput Library Transgenesis in *Caenorhabditis elegans* via Transgenic Arrays Resulting in Diversity of Integrated Sequences (TARDIS) | https://www.ncbi.nlm.nih.gov/bioproject/?term=PRJNA893002 | NCBI BioProject, PRJNA893002 |

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
