## [Editor Report · eLife assessment]

This manuscript provides a description of an approach for efficiently integrating diverse libraries into the *C. elegans* genome and tools that enable researchers to use the method. It is a **valuable** contribution for researchers carrying out experiments that would benefit from easy generation of such libraries, and the data for the effectiveness of the method is **solid**. The advantages of this approach in terms of ease and effectiveness relative to others with similar aims will emerge as they are put to more general use in addressing biological problems.

---

## [Referee Report · Reviewer #1 (Public Review)]

This work describes a novel high-throughput approach to diverse transgenesis which the authors have named TARDIS for Transgenic Arrays Resulting in Diversity of Integrated Sequences. The authors describe the general approach: the generation of a synthetic 'landing pad' for transgene insertion (as previously reported by this group) that has a split selection hygromycin resistance gene, meaning that only perfect integration with the insert confers resistance to the otherwise lethal hygromycin drug. The authors then demonstrate two possible applications of the technology: individually barcoded lineages for lineage tracing and transcriptional reporter lines generated by inserting multiple promoters. In both cases, the authors did a limited 'proof of concept' study including many important controls, showcasing the potential of the method. The conclusions for applications of this method in *C. elegans* are supported by the data and the authors discuss important observations and considerations. In the discussion, the discuss the potential application of the method beyond *C. elegans*, although this remains speculative at this point given that a nontrivial aspect of the success of the method in worms is the self-assembly of DNA into heritable extrachromosomal arrays (a feature that is absent in most other systems).

---

## [Referee Report · Reviewer #2 (Public Review)]

This paper explores the possibility of integrating diverse and multiple DNA fragments in the genome taking advantage of plasmids in arrays, and CRISPR. Since the efficiency of integration in the genome is low, they, as others in the field, use selection markers to identify successful events of integration. The use of these selection markers is common and diverse, but they use a couple of distinct strategies of selection to:

- Introduce bar codes in the genome of individuals at one specific genomic site (gene for Hygromycin resistance with bar code in an intron with homology arms to complete a functional gene);

- Introduce promoters at two specific genomic landing pads downstream of fluorescent reporters.

The strengths of the study are the clever design of the selection markers, which enrich the collection of this type of markers. While the work is not methodologically novel - it adds to other recent studies, e.g. from Nonet, Mouridi et al., and Malaiwong et al, that use the integration of single and multiple/diverse DNA sequences in the *C. elegans* genome - it provides a protocol for doing so and tool to make it practical. A limited number of experiments using the method are presented here, and the real test of this method will be its use to address biological questions.

---

## [Author Response]

The following is the authors' response to the original reviews.

**Consolidated response to public comments:**

We are grateful to the reviewers for their careful examination of our manuscript and for their insights for improving our work. We appreciate that they recognize the potential of the TARDIS approach for diverse transgenesis applications.

We address two primary concerns that the reviewers raise. First is a concern that this approach is not as innovative as stated. We acknowledge that our work builds upon previous studies in the field, such as those by Nonet, Mouridi et al., with Malaiwong coming after our initial preprint. However, we believe that our approach offers a unique contribution, in that prior work does not provide a protocol or process to provide large-scale multiplexed transgenesis. Specifically, our introduction of large sequence library arrays (TARDIS Library Arrays or TLAs). While high throughput multiplexed transgenesis is discussed in Nonet & Mouridi manuscripts, it is never demonstrated. It is the combination of library construction, heritable transmission of the library itself, and then induced transgenesis of library components at a defined location within single individuals that makes this approach particularly useful.

Second, there were concerns that we have not demonstrated that this approach will work beyond *C. elegans*. We agree that our discussion of the potential application of TARDIS beyond *C. elegans* is speculative at this point. Our intention was to highlight the potential for future development and application in other systems. In some cases, large integrations into the genome are possible, such as in the case of H11 locus in mice, which could provide a means to inherit a sequence library. We are hopeful that our success in *C. elegans* will inspire work in other systems. The motivation for this will naturally depend on the usefulness of actual TARDIS implementations, which will be forthcoming in due course.

**Reviewer #1 (Recommendations For The Authors):**
1. Section titled "Integration from TARDIS array to F1" beginning on line 161 has some missing details that make it difficult to follow. Many of those details are present in the following section titled "Generation and Integration of TARDIS promoter library", but should have been present sooner.a. How many barcodes were in the array in line PX786?b. Clarify the use of G-418, heat shock, hygromycin, etc. in this paragraph.c. Please clarify that the L1 death is due to selection with G-418 - "We found that a portion of the initially plated worms die, likely due to lack of array inheritance." is confusing unless you add that they are selected in this step.d. "These results suggest that approx. 100-200 worms need to be heat shocked to obtain an integrated line" - the math actually looks like 200-300, and this would be to get a single integrant.2. In general, the barcoding study and results reported here read like a teaser/proof-of-concept but do not really robustly demonstrate the application of the method for barcoding and tracing individual lineages in a population of *C. elegans*. How many barcodes were in the array, and how many ended up in F1s? Would one need to screen for duplicate barcodes after integration?3. The promoter library study is impressive but again, rather limited.4. The Discussion section about extending this technology to other systems is fairly balanced, acknowledging the limitations that would need to be overcome. The language in the abstract and introduction is less balanced and oversells the current translation of this approach to systems outside *C. elegans*.
**Reviewer #2 (Recommendations For The Authors):**
As I mentioned in the Public Review, I appreciate the design of the selection markers for integration. However, I do not see a major advance in the field. The use of barcoding of individuals to address a biological question would change that impression.Regarding the integration of promoters, I think this is something that anyone could address in diverse forms using existing knowledge.Suggestions:- Use one or two more landing pads for barcoding of animals and check numbers, efficacy, enrichments..etc. About 500 sequences overrepresented may be too much for future applications;- Increase the number of landing pads for inserting promoters. Genomics context matters and this could help to have a better summary of the real expression patterns driven by the promoter of interest;- Other references about landing pads would be Vicencio et al, Genetics 2019, and Nonet microPublication Biology 2021.

In addition to the general comments, the reviewers provided useful suggestions to the text that we have used to clarify the manuscript.